# The selective PGI$_2$ receptor agonist selexipag ameliorates Sugen 5416/hypoxia-induced pulmonary arterial hypertension in rats

Yohei Honda[1]*, Keiji Kosugi[2], Chiaki Fuchikami[1], Kazuya Kuramoto[2], Yuki Numakura[1], Keiichi Kuwano[2]

1 Discovery Research Laboratories, Nippon Shinyaku Co., Ltd, Kyoto, Japan, 2 R&D Administration Division, Nippon Shinyaku Co., Ltd, Kyoto, Japan

* y.honda@po.nippon-shinyaku.co.jp

**Data Availability Statement:** All relevant data are within the paper and its Supporting Information files.

## Abstract

Pulmonary arterial hypertension (PAH) is a lethal disease characterized by a progressive increase in pulmonary artery pressure due to an increase in vessel tone and occlusion of vessels. The endogenous vasodilator prostacyclin and its analogs are used as therapeutic agents for PAH. However, their pharmacological effects on occlusive vascular remodeling have not been elucidated yet. Selexipag is a recently approved, orally available and selective prostacyclin receptor agonist with a non-prostanoid structure. In this study, we investigated the pharmacological effects of selexipag on the pathology of chronic severe PAH in Sprague-Dawley and Fischer rat models in which PAH was induced by a combination of injection with the vascular endothelial growth factor receptor antagonist Sugen 5416 and exposure to hypoxia (SuHx). Oral administration of selexipag for three weeks significantly improved right ventricular systolic pressure and right ventricular (RV) hypertrophy in Sprague-Dawley SuHx rats. Selexipag attenuated the proportion of lung vessels with occlusive lesions and the medial wall thickness of lung arteries, corresponding to decreased numbers of Ki-67-positive cells and a reduced expression of collagen type 1 in remodeled vessels. Administration of selexipag to Fischer rats with SuHx-induced PAH reduced RV hypertrophy and mortality caused by RV failure. These effects were probably based on the potent prostacyclin receptor agonistic effect of selexipag on pulmonary vessels. Selexipag has been approved and is used in the clinical treatment of PAH worldwide. It is thought that these beneficial effects of prostacyclin receptor agonists on multiple aspects of PAH pathology contribute to the clinical outcomes in patients with PAH.

## Introduction

Pulmonary arterial hypertension (PAH) is a lethal disease characterized by a progressive increase in pulmonary artery pressure, which causes right ventricular (RV) hypertrophy leading to right heart failure [1]. The increase in pulmonary artery pressure is caused by an

**Funding:** All authors are employed by Nippon Shinyaku Co., Ltd. The funders had no role in study design, data collection and analysis, or preparation of the manuscript.

**Competing interests:** All authors are employed by Nippon Shinyaku Co., Ltd. The funders had no role in study design, data collection and analysis, or preparation of the manuscript. This does not alter our adherence to PLOS ONE policies on sharing data and materials.

imbalance in the regulation of vessel tone and occlusion of the vessels by excessive proliferation of pulmonary arterial smooth muscle cells in the lumen [2].

Prostacyclin is one of the most important endogenous pulmonary vasodilators. The prostacyclin receptor plays an important role in the relaxation of vessel tone and in the regulation of the proliferation of pulmonary arterial smooth muscle cells. Therefore, prostacyclin and its analogs are widely used as therapeutic agents for the treatment of PAH [3–5]. Selexipag is a recently developed orally available and selective prostacyclin receptor agonist with a non-prostanoid structure [6]. In the phase III trial GRIPHON, the selexipag treatment group showed a significant reduction in the risk of the primary composite endpoint of death or a complication related to PAH compared with placebo [7]. Selexipag has been approved for the treatment of adult patients with PAH in the US, the EU and Japan [8].

In spite of the great success of prostacyclin receptor agonists as therapeutic agents for PAH, their pharmacological effects on the pathology of PAH, such as complex occlusive vascular remodeling and mortality risk associated with RV failure, have not yet been elucidated. The Sprague-Dawley (SD) rat model in which PAH is induced by a combination of subcutaneous injection of a vascular endothelial growth factor inhibitor, Sugen 5416, and exposure to hypoxia (SuHx) can reproduce PAH-specific complex occlusive vascular remodeling such as plexiform lesions [9,10]. This pathological feature is not observed in other commonly used PAH animal models which are induced by administration of monocrotaline or by only exposure to hypoxia [11]. The SD SuHx model also develops severe RV hypertrophy, systolic and diastolic RV dysfunction, and increased glucose uptake in the RV [12]. These progressive RV failure phenotypes closely resemble those of patients with PAH. In consideration of these facts, the SD SuHx model is regarded as one of the best rodent models for studying PAH. Recently some groups reported that the phenotypes induced by SuHx vary among rat strains. For example, in Wistar Kyoto rats SuHx induces not only occlusive vascular remodeling but also emphysema, whereas in SD rats SuHx induces only occlusive vascular remodeling [13,14]. In Fischer rats, SuHx induces severe RV failure similar to those induced in SD rats, but whereas SD SuHx rats show good survival SuHx Fischer rats tend to die of the failure of RV adaptation to increasing afterload. For these reasons, we chose the SD SuHx rat model to assess the effect of selexipag on hemodynamics and occlusive vascular remodeling and the Fischer SuHx rat model to assess its effect on mortality by RV failure.

The efficacy of the prostacyclin receptor agonists iloprost and treprostinil have been assessed in the SD SuHx model. They improve the hemodynamics but do not show ameliorative effects on pulmonary vascular remodeling [15,16]. We have previously reported that MRE-269, an active metabolite of selexipag, has potent vasodilator effects on lung vessels compared with other prostacyclin receptor agonists due to its high selectivity [17]. We hypothesize that selexipag can cause potent vasodilation even in the severe PAH of the SuHx model and improve the pathological features of PAH, so in the present study we investigated the pharmacological effects of selexipag on the pathological features of PAH in the SD SuHx rat model and mortality by RV failure in the Fischer SuHx rat model.

## Materials and Methods

### Animals

All animal procedures were approved by the Committee for the Institutional Care and Use of Animals of Nippon Shinyaku Co., Ltd., which are based on the Law for the Humane Treatment and Management of Animals (Law No. 105, 1 October 1973, as revised on 1 June 2006). Six-week-old male SD rats (Japan SLC, Shizuoka, Japan) were used for all experiments except the survival experiment, in which six-week-old male F344/DuCrlCrlj (Fischer) rats (Japan

Charles River, Yokohama, Japan) were used. The animals were housed three per cage in a room maintained at 20–26°C and 35–75% relative humidity with an alternating 12-h light/dark cycle (the lights came on automatically at 8:00 a.m.) and allowed free access to pellet chow (F-2; Funabashi Farm, Chiba, Japan) and tap water. Animals were allocated to groups by their body weights the day before the start of treatment in each experiment. The animals were well cared for by the animal care personnel and by the veterinarian at our institution. The condition of the animals was checked twice a day. Humane endpoints were used during all experiments. If any animals had shown either (1) persistent crouching or noisy breathing or (2) a loss of body weight greater than 20%, they would have been euthanized by the use of carbon dioxide to avoid suffering. However, no animals met either of those criteria during any of the experiments.

## U46619-induced elevation of right ventricular systolic pressure (RVSP)

Rats were anesthetized by subcutaneous injection of ethyl carbamate (Wako, Osaka, Japan) at 1.2 g/kg. The rat was laid on its back and its temperature was maintained at 37°C with an animal blanket controller (model ATB-1100; Nihon Kohden, Tokyo, Japan) throughout the measurement. A polyethylene cannula (PE-50; Becton, Dickinson and Company, Franklin Lakes, NJ) filled with heparinized saline was inserted into the right femoral artery for measurement of the mean arterial pressure (MAP) and heart rate (HR) and another was inserted into the right femoral vein for intravenous administration of U46619 (Cayman Chemical, Ann Arbor, MI), which was dissolved in dimethyl sulfoxide (5 mg/mL) and diluted with saline (60 μg/mL). Another cannula was inserted into the right jugular vein for measuring the RVSP. For administration of selexipag, a silicon tube was inserted into the duodenum. Selexipag (synthesized by Nippon Shinyaku Co., Ltd.) was suspended in 0.5% (w/v) methylcellulose (Metolose SM-400; Shin-Etsu Chemical, Tokyo, Japan) in distilled water (Otsuka Pharmaceutical, Tokushima, Japan). Acute elevation of RVSP was produced by continuous infusion of U46619. The infusion rate was adjusted within the range of 0.6–1.2 μg/kg/min to give an RVSP of 45–55 mmHg. After the RVSP had stabilized, the value of RVSP at 0 min was recorded and selexipag (3 mg/kg) or vehicle (0.5% methylcellulose solution) was administered intraduodenally (N = 8 per group). RVSP was measured and recorded for 120 min after administration of selexipag. At 10, 20, 30, 60, 90 and 120 min after administration, the 1-min average values of RVSP were calculated with the PowerLab software (ADInstruments, Bella Vista, Australia) and used as the measured values. The change in RVSP from its value at 0 min was calculated for the measurements from 10 to 120 min.

## SuHx-induced PAH model

Sugen 5416 was synthesized by Nippon Shinyaku Co., Ltd., and suspended at 5 mg/mL in a buffer containing 0.5% (w/v) carboxymethyl cellulose sodium salt (Sigma-Aldrich, St. Louis, MO), 0.9% (w/v) sodium chloride (Nacalai Tesque, Kyoto, Japan), 0.4% (v/v) polysorbate 80 (Sigma-Aldrich) and 0.9% (v/v) benzyl alcohol (Nacalai Tesque). Six-week-old SD rats were subcutaneously injected with Sugen 5416 (20 mg/kg) and exposed to 10% oxygen for three weeks (SuHx) [18]. Thereafter they were returned to normoxia and maintained on normoxia. Normoxia rats were kept in room air during the experimental period. Vehicle-treated groups were orally administered 0.5% methylcellulose solution twice daily. Selexipag-treated groups were orally administered compound suspended in 0.5% methylcellulose solution as two daily injections. The treatment protocol for each study group is shown in Fig 2. In the early-stage study, SuHx rats were administered either selexipag (10 or 30 mg/kg) or vehicle for three weeks (from 3 weeks after Sugen 5416 injection to the end of the 6th week), then sacrificed to

measure the outcomes (Fig 2A; N = 8 for the normoxia group and N = 10 for the other groups). In the late-stage study, SuHx rats were administered either selexipag (30 mg/kg) or vehicle for three weeks (from 5 weeks after Sugen 5416 injection to the end of the 8th week), then sacrificed to measure the outcomes (Fig 2B; N = 10 per group). The SuHx rats which were sacrificed at the beginning of the treatment were used as pre-treatment control groups. RVSP and RV hypertrophy were measured as outcomes 24 h after the final administration of selexipag. The left lungs of the rats were collected at the time of measurement of the outcomes and used for histological analysis.

## Measurement of arterial pressure and heart rate

The day before RVSP evaluation, the mean arterial pressure (MAP) and heart rate (HR) were measured with a tail-cuff indirect blood pressure meter (BP-98A; Softron, Tokyo, Japan). The conscious rat was placed in a holder maintained at 37°C and the cuff attached to its tail, and then the MAP and HR were measured.

## RVSP measurements

Rats were anesthetized with isoflurane (Pfizer Inc., New York, NY) and mechanically ventilated (Dwyer SAR-830/P ventilator; CWE, Ardmore, PA). The rat was laid on its back and its body temperature was maintained at 37°C with an animal blanket controller throughout the experimental period. A polyethylene cannula filled with heparinized saline was inserted into the right jugular vein and advanced into the right ventricle for monitoring the RVSP. The cannula was connected to a non-isolated bridge amplifier (FE228 Octal Bridge Amp; ADInstruments), and the RVSP was recorded with a PowerLab 16/35 data acquisition system (ADInstruments). The system was carefully calibrated with a mercury manometer (201-50-22; Acoma, Tokyo, Japan) before each experiment.

## Measurement of RV hypertrophy and tissue preparation

After RVSP evaluation, rats were euthanized by exsanguination from the abdominal aorta under 3% isoflurane anesthesia. The left lungs were removed, perfused with heparinized saline and then fixed in 10% neutral buffered formalin. The right ventricle (RV) and left ventricle plus septum (LV+S) were weighed separately and the ratio of RV to LV+S [RV/(LV+S)] was calculated.

## Histological analysis

The embedding of the left lungs in paraffin and tissue staining were performed by Applied Medical Research Laboratory (Osaka, Japan). The stained sections were digitized with an Aperio CS2 Digital Pathology Scanner (Leica Biosystems, Nussloch, Germany) and analyzed with the Aperio ImageScope software (Leica Biosystems). To evaluate the occlusive vessel lesions, including medial wall hypertrophy, reaction/proliferation of the vascular endothelium, and concentric laminar neointimal lesions, small vessels whose outer diameters ranged from 30 to 50 μm (30 vessels per section) were assessed. Specifically, the occlusive ratio of the lumen was determined from the vessel outer diameter and the luminal diameter. An occlusive vessel lesion was defined as a partial (>50%) or full obstruction of the lumen of a vessel [10,19]. For morphometric analysis, the medial wall thickness was assessed in arteries whose outer diameters ranged from 50 to 200 μm. The percent medial thickness of arteries was calculated with the formula [(outer diameter—inner diameter)/outer diameter] × 100 [10]. For immunohistochemical analysis, paraffin-embedded lung tissue slides were deparaffinized and hydrated.

Sections were boiled in ethylenediaminetetraacetic acid buffer (pH 9.0) for 40 min to retrieve the Ki-67 epitope or incubated with proteinase K for 5 min at room temperature to retrieve the von Willebrand factor (vWF) or collagen type I epitope. After quenching endogenous peroxidase activity with 0.3% hydrogen peroxide in methanol, the sections were incubated with anti-Ki-67 antibody (diluted 1:1; Nichirei Biosciences, Tokyo, Japan) for 60 min at room temperature, anti-vWF (diluted 1:5000; Nichirei Biosciences) overnight at 4˚C or anti-collagen type I antibody (diluted 1:100; Southern Biotechnology Associates, Birmingham, AL) overnight at 4˚C. Immunoreactivity was visualized using 3,3'-diaminobenzidine (DAB). Apoptotic cells in the lung vessels were detected by terminal deoxynucleotidyl transferase mediated 2'-deoxyuridine 5'-triphosphate nick-end labeling (TUNEL) assay using the ApopTag Peroxidase *In Situ* Apoptosis Detection Kit (Merck, Darmstadt, Germany). The small vessels (outer diameter, 30–50 μm; 30 vessels per section) were assessed for the proportion of Ki-67-positive vessels and the expression of collagen type I. Vessels containing more than one cell stained with anti-Ki-67 antibody were counted as Ki-67-positive vessels. To evaluate the expression level of collagen type I, the vessel area was determined as the area encircled by the tunica adventitia and DAB-positive pixels in the vessel area were counted with Positive Pixel Count v9, a macro in the ImageScope software. Then the ratio of the number of positive pixels to the vessel area was calculated for each vessel and the average number of positive pixels per square micrometer for 30 vessels in each lung was used as the value for the individual section for statistical analysis [10,20,21].

## Mortality and RV hypertrophy in SuHx-induced PAH Fischer rats

Six-week-old Fischer rats were subcutaneously injected with Sugen 5416 (20 mg/kg) and exposed to 10% oxygen for three weeks, after which they were returned to normoxia. The rats were treated with vehicle (0.5% methylcellulose solution) or selexipag (30 mg/kg) orally twice daily from day 22 to day 42 after Sugen 5416 subcutaneous injection (N = 10 per group). The general condition of the rats was checked twice daily. Observation was terminated on day 42 and all surviving animals were euthanized by exsanguination from the abdominal aorta under 3% isoflurane anesthesia to measure their RV hypertrophy.

## Statistical analysis

All data were expressed as the mean ± S.E.M. and figures were drawn with GraphPad Prism 6 (GraphPad Software, San Diego, CA). In the evaluation of the U46619-induced elevation of RVSP, the selexipag-treated group was compared with the vehicle-treated group by repeated measures analysis of variance (ANOVA). When a significant difference was noted, the significant differences at each time point were analyzed by Student's *t*-test between the two groups. In the SuHx models, statistical analysis was performed by Tukey's test. Kaplan-Meier curves for survival rates were drawn with GraphPad Prism 6 and compared using log-rank analysis. RV hypertrophy in the SuHx Fischer model was analyzed by Student's *t*-test. All statistical analyses were performed with SAS System Version 9.3 (SAS Institute Inc., Cary, NC) and EXSUS Version 8.1.0 (CAC Croit Corporation, Tokyo, Japan). A P value of less than 0.05 was considered statistically significant.

## Results

### Effect of selexipag on U46619-induced elevation of RVSP

Prior to investigating the effects of selexipag on the pathology of PAH, we verified the vasodilating activity of selexipag in lung and systemic vessels in an acute rat model of PAH induced

by the thromboxane $A_2$ agonist U46619. Thromboxane $A_2$ is produced by activated platelets and induces platelet aggregation and vasoconstriction [22]. Its plasma levels are increased in patients with PAH, and activation of the thromboxane receptor on vascular smooth muscle cells leads to vasocontraction through the accumulation of inositol phosphates [23,24]. Accordingly, intravenous infusion of a thromboxane agonist, U46619, to rats quickly induces elevation of RVSP and this is used to create an acute PAH model [25,26]. Intravenous infusion of U46619 produced an elevation in RVSP from 35.5 ± 0.9 to 48.1 ± 0.7 mmHg. We have previously reported that the administration of 3 mg/kg selexipag causes a slightly increased heart rate but has no effect on systemic blood pressure in rats not infused with U46619 [27]; therefore we chose a dose of 3 mg/kg to investigate the effect of selexipag on RVSP. This elevation of RVSP was significantly reduced within 30 min by intraduodenal administration of selexipag at 3 mg/kg, and the effect persisted until the end of measurement at 120 min (Fig 1A). No significant effect of selexipag on the mean arterial pressure (MAP) or heart rate (HR) was observed at any time during the experiments (Fig 1B and 1C). The maximum reduction in RVSP was approximately 16% (7.9 mmHg).

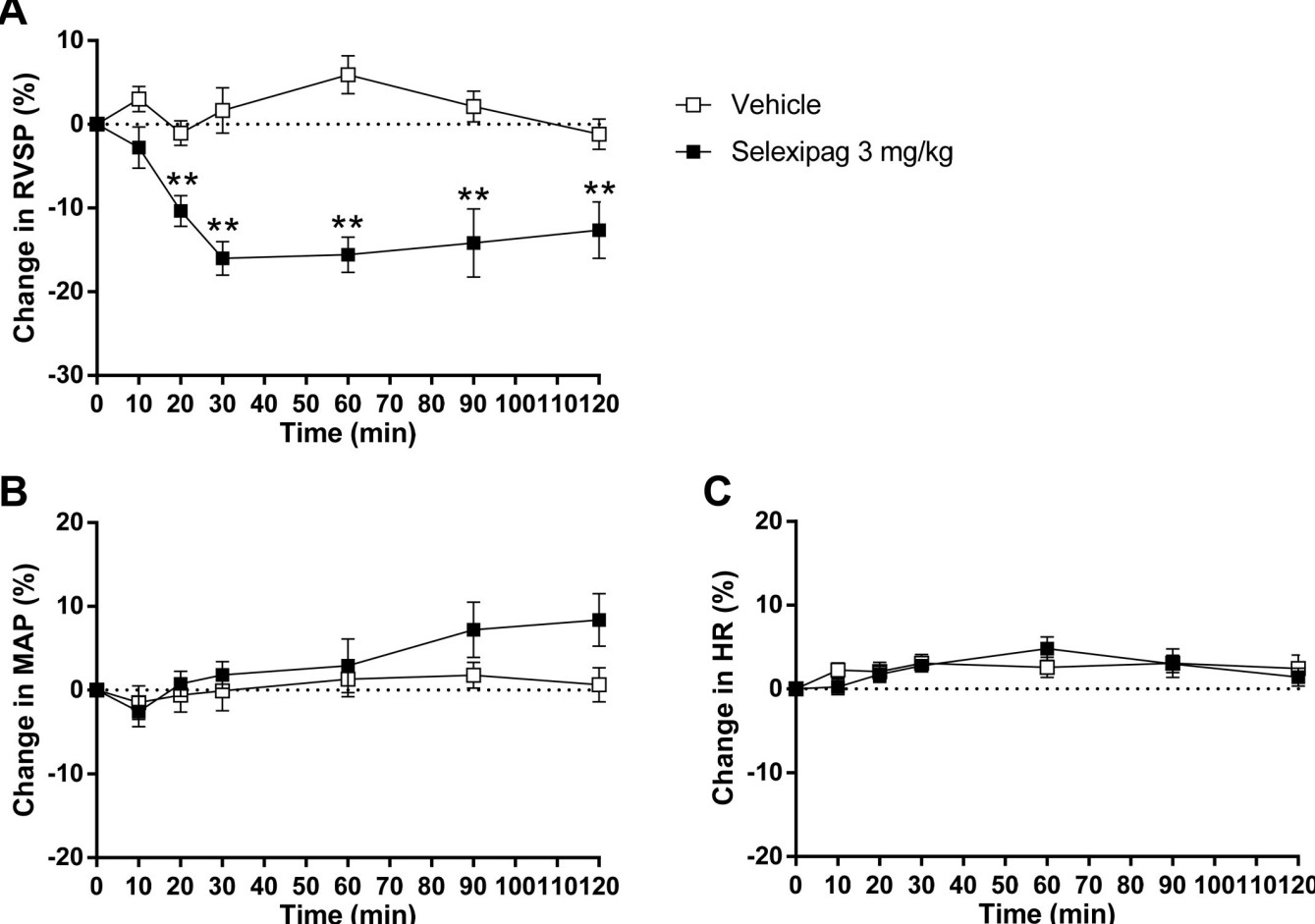

**Fig 1. Hemodynamic effects of selexipag in U46619-induced acute PAH model rats.** Effects of selexipag on right ventricular systolic pressure (RVSP) (A), mean arterial blood pressure (MAP) (B) and heart rate (HR) (C). The percentage change in RVSP, MAP and HR from the value at 0 min were calculated for each measurement until 120 min. Statistical analyses were performed using repeated measures ANOVA followed by Student's *t*-test. ** $P < 0.01$ vs. vehicle. Values are means ± S.E.M. N = 8 per group.

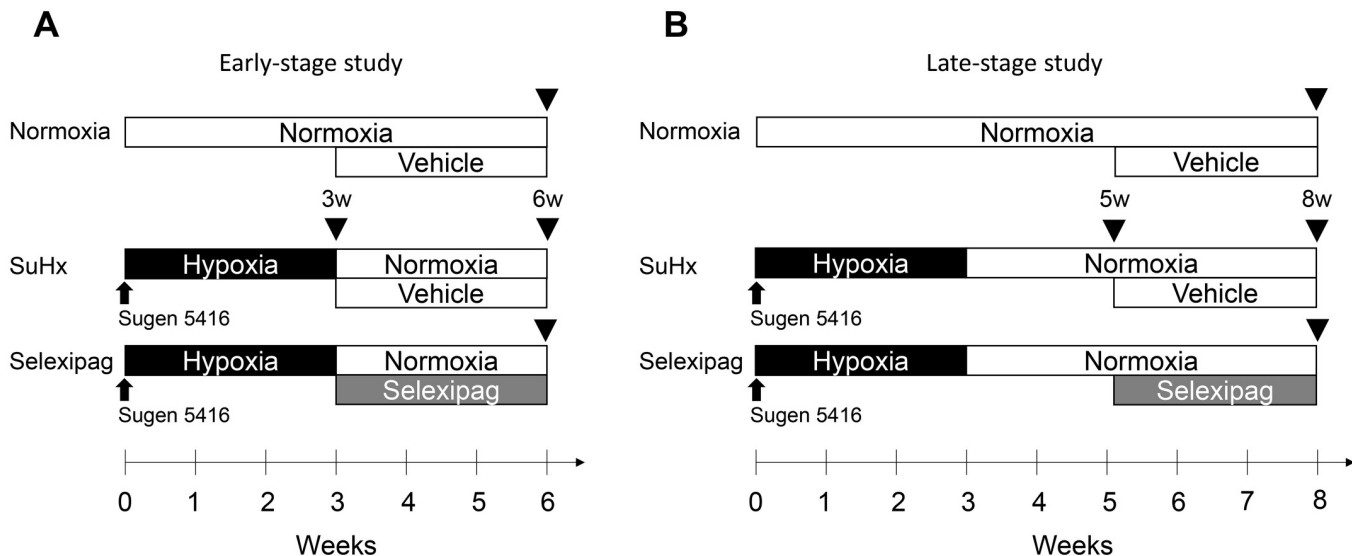

**Fig 2. Study design using the SuHx rat model.** Scheme of the early-stage study (A) and the late-stage study (B). Black arrowheads indicate the date for the assessment of the outcome parameters.

## Effects of selexipag on hemodynamic parameters, RV hypertrophy and obstructive pulmonary vascular remodeling in the early phase of SuHx rats

To investigate the pharmacological effects of selexipag on the pathology of PAH, we tested selexipag on both the early-pathology and the late-pathology phase in SuHx rats (Fig 2).

Subcutaneous injection of SD rats with Sugen 5416 followed by exposure to 10% oxygen for three weeks induced a significant increase in RVSP (from $37.6 \pm 0.8$ to $95.7 \pm 4.2$ mmHg; Fig 3A) and RV/(LV+S) (from $0.23 \pm 0.01$ to $0.63 \pm 0.02$; Fig 3B). The severity of PAH differs between the acute U46619-induced and the chronic SuHx-induced rat models. The increase in RVSP in the SuHx model (from $37.6 \pm 0.8$ to $141.1 \pm 6.2$ mmHg for SuHx 6w; Fig 3A) was much greater than in the acute model (from $35.5 \pm 0.9$ to $48.1 \pm 0.7$ mmHg). Therefore, the fact that the higher dose of 10 or 30 mg/kg/day selexipag was needed in our SuHx model compared with the single administration of 3 mg/kg in the acute model may have been due to the more serious development of RVSP in the SuHx model. For the experiment on the early-pathology phase, selexipag (10 or 30 mg/kg/day) or vehicle was administered orally for three weeks. At the end of the treatment, the RVSP of the vehicle-treated group (SuHx 6w) had further significantly increased (to $141.1 \pm 6.2$ mmHg), while the ratio RV/(LV+S) showed no further increase. In contrast, in the group treated with selexipag at 30 mg/kg, although RVSP still remained high, it significantly decreased to $102.1 \pm 5.7$ mmHg with no effect on MAP or HR (Fig 3A, 3C and 3D). In concert with the decrease in RVSP, RV/(LV+S) was significantly reduced by both doses of selexipag relative to that of the vehicle-treated group (to $0.56 \pm 0.02$ at 10 mg/kg and $0.50 \pm 0.02$ at 30 mg/kg; Fig 3B). Occlusive lesion in pulmonary vessels is observed in patients with PAH and in SuHx rats, and it causes an increase in pulmonary vascular resistance and an elevation of RVSP [1,18]. To investigate the effect of selexipag on occlusive lesion, we histologically evaluated the proportion of small vessels with occlusive lesions in the lungs of SuHx rats. The rate of occlusive lesions was $14 \pm 2\%$ of the vessels at the end of hypoxia (SuHx 3w), and it progressed to $55 \pm 5\%$ at six weeks after Sugen 5416 injection in the vehicle-treated group (SuHx 6w) (Fig 4A and 4B). The progression of occlusive lesions

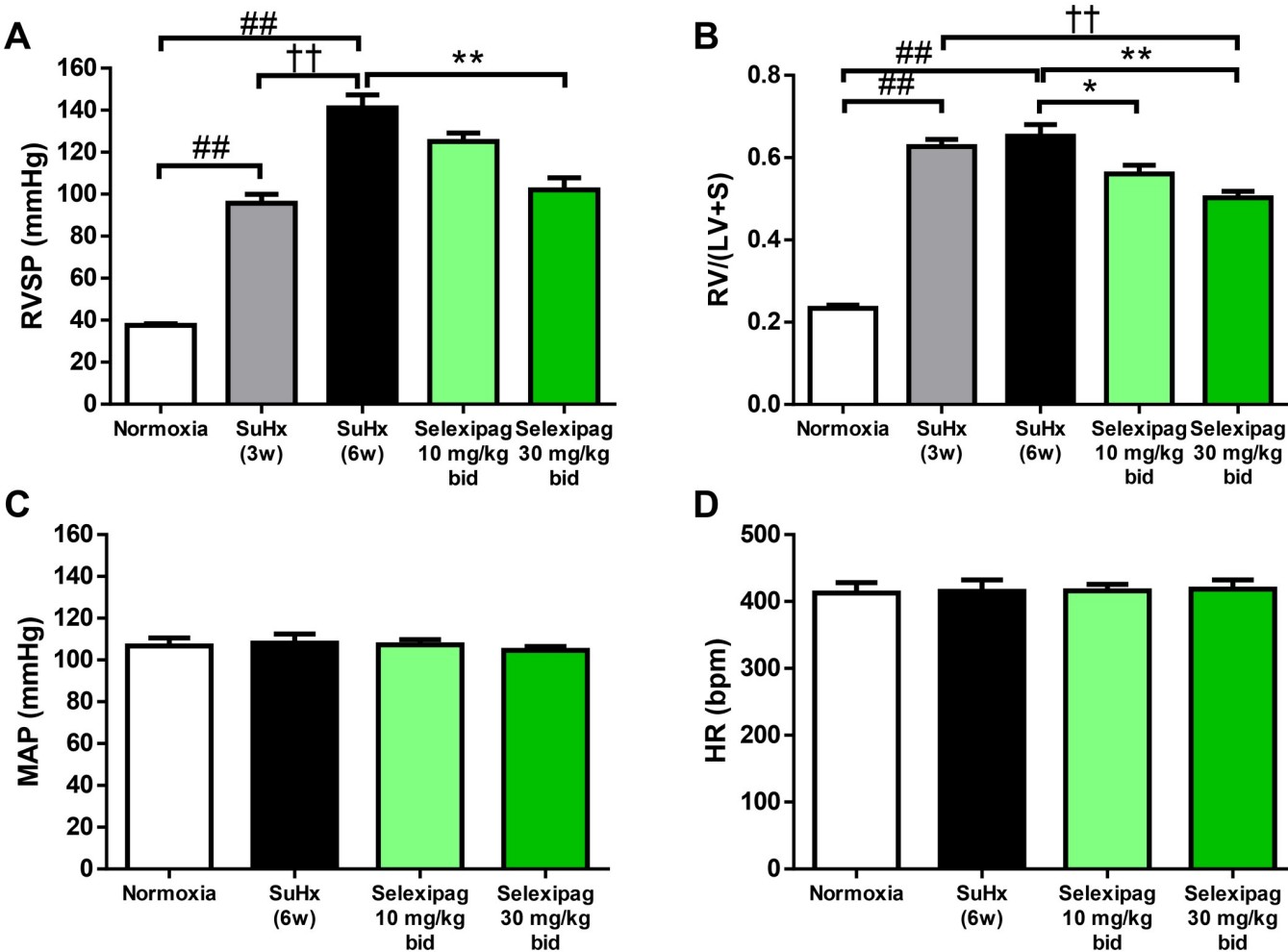

**Fig 3. Hemodynamic effects of selexipag in the early stage of SuHx rats.** Effect of selexipag on right ventricular systolic pressure (RVSP) (A), the ratio of the weight of the right ventricle to that of the left ventricle + septum (RV/LV+S) (B), mean arterial blood pressure (MAP) (C) and heart rate (HR) (D). Statistical analyses were performed using Tukey's test. ##$P<0.01$ vs. normoxia, ††$P<0.01$ vs. SuHx-3w, *$P<0.05$ and **$P<0.01$ vs. SuHx-6w. Values are means ± S.E.M. N = 8–10 per group.

corresponds to that of the elevation of RVSP. The rate of occlusive lesions was significantly reduced to 26 ± 7% in the group treated with 30 mg/kg selexipag compared with the vehicle-treated group, corresponding to the decrease in RVSP.

## Effects of selexipag on hemodynamic parameters, RV hypertrophy and obstructive pulmonary vascular remodeling in the late stage of SuHx rats

For the experiment on the early-pathology phase, selexipag at 30 mg/kg, but not 10 mg/kg, significantly improved RVSP and occlusive lesions. Therefore, in the next step, we used selexipag at 30 mg/kg to investigate the late-pathology phase. For the experiment on the late-pathology phase, SuHx rats were maintained under normoxic conditions for two weeks after exposure to 10% oxygen and then administered selexipag or vehicle for three weeks. Both RVSP and RV/(LV+S) were significantly elevated at the start of treatment (SuHx 5w; from 37.9 ± 0.8 to 116.2 ± 5.1 mmHg for RVSP and from 0.23 ± 0.01 to 0.62 ± 0.03 for RV/(LV+S)) and showed

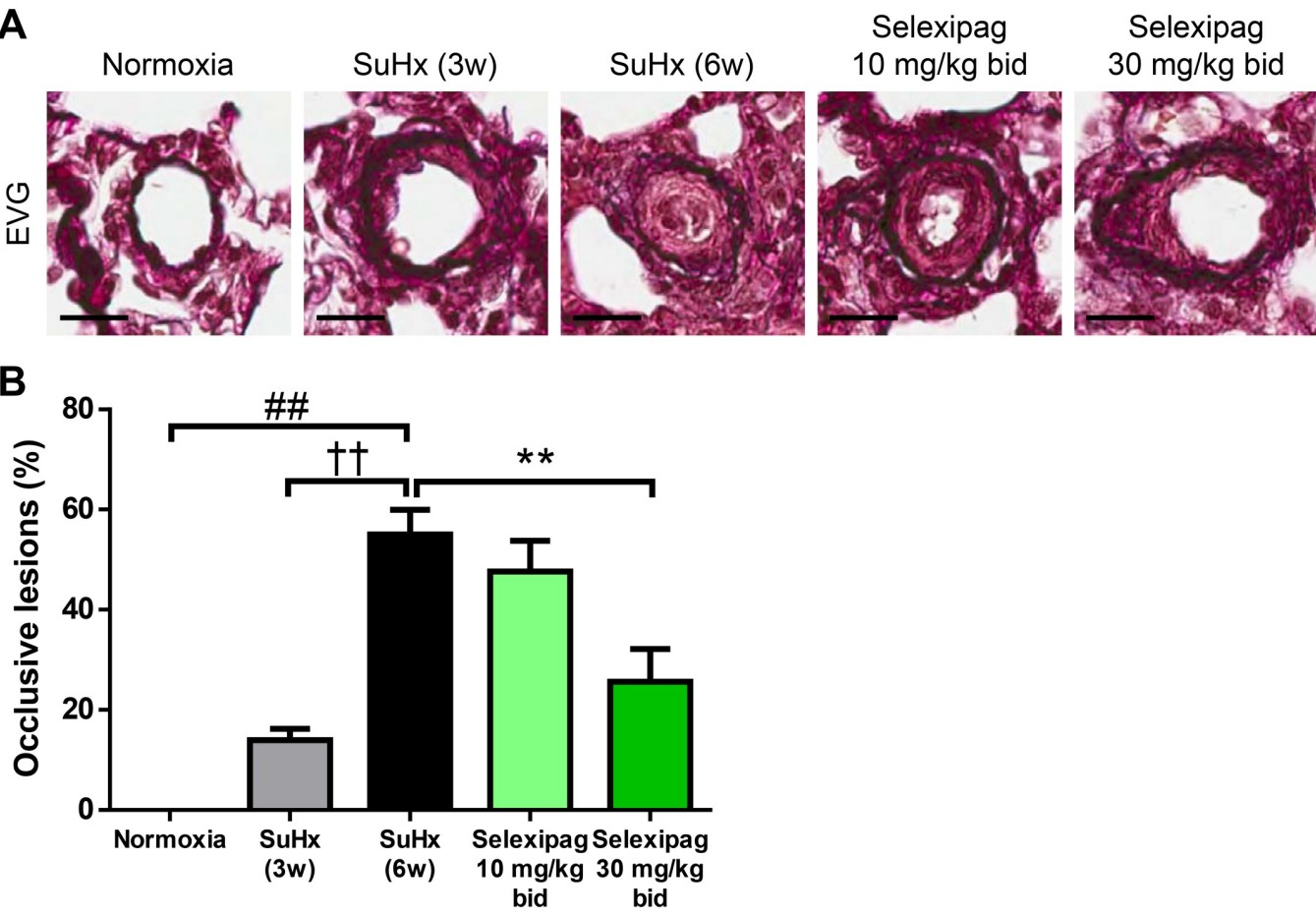

**Fig 4. Effect of selexipag on occlusive lesions in the early stage of SuHx rats.** Representative photographs of occlusive lesions of pulmonary vessels (scale bars = 20 μm) (A) and the percentage of occlusive lesions among the vessels (B). Statistical analysis was performed using Tukey's test. ##$P$<0.01 vs. normoxia, ††$P$<0.01 vs. SuHx-3w, **$P$<0.01 vs. SuHx-6w. Values are means ± S.E.M. N = 8–10 per group.

no further increase in the vehicle-treated group (SuHx 8w; 114.6 ± 7.3 mmHg for RVSP and 0.63 ± 0.03 for RV/(LV+S)) (Fig 5A and 5B). In the selexipag-treated group, RVSP (82.1 ± 8.1 mmHg) and RV/(LV+S) (0.43 ± 0.02) were significantly decreased not only compared with the vehicle-treated group (SuHx 8w) but also compared with the pre-treatment group (SuHx 5w). No effect on MAP or HR was observed (Fig 5C and 5D). Similar to the elevation of RVSP, the rate of occlusive lesions had significantly increased to 41 ± 5% five weeks after Sugen 5416 injection in the pre-treatment group (SuHx 5w) and showed no further significant change at eight weeks in the vehicle-treated group (SuHx 8w; 54 ± 5%) (Fig 6A and 6B). The rate of occlusive lesions was significantly reduced to 28 ± 5% in the group treated with 30 mg/kg selexipag not only compared with the vehicle-treated group but also compared with the pre-treatment group. An elevation of RVSP propagates an elevation of pulmonary artery pressure and leads to thickening of the medial wall of the pulmonary arteries in PAH patients [28]. The medial walls of pulmonary arteries were thickened in SuHx rats (normoxia, 12.2 ± 1.1%; SuHx 5w, 27.4 ± 1.7%; SuHx 8w, 29.5 ± 1.6%) and the thickening was significantly attenuated in the selexipag-treated group (16 ± 1.1%) compared with both the pre-treatment group and the vehicle-treated group (Fig 6A and 6C).

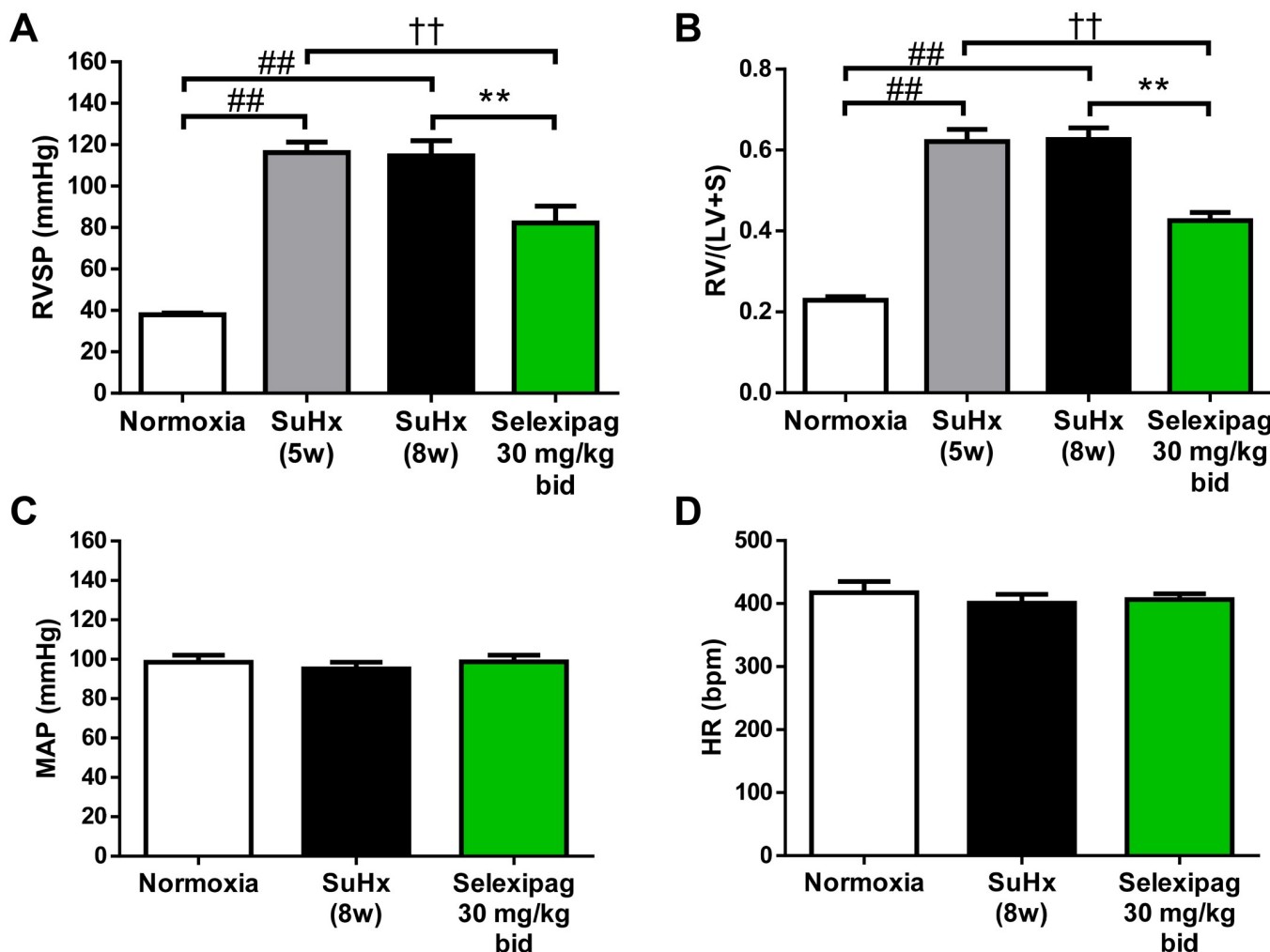

**Fig 5. Hemodynamic effects of selexipag in the late stage of SuHx rats.** RVSP (A), RV/LV+S (B), mean arterial blood pressure (MAP) (C) and heart rate (HR) (D). Statistical analyses were performed using Tukey's test. ##$P<0.01$ vs. normoxia, ††$P<0.01$ vs. SuHx-5w, **$P<0.01$ vs. SuHx-8w. Values are means ± S.E.M. N = 10 per group.

### Antiproliferative and apoptosis-inducing effect of selexipag in occluded small lung vessels from the late stage in SuHx rats

To investigate the detailed mechanism of the protective action of selexipag on PAH, we used lung vessels obtained from the late-pathology phase, because we presumed that it reflected the clinical setting in which selexipag is administered after establishment of the disease. Vascular occlusive lesions in the lungs of PAH patients are consistent with abnormal cell proliferation and vessel fibrosis [28,29]. An increase in cells positive for the proliferating cell marker Ki-67 and the expression of fibrotic proteins, including collagen type I, are observed in occluded lung vessels of SuHx rats [10,20]. The proportion of lung vessels with Ki-67-positive cells was 41 ± 2.9% in the pre-treatment group (SuHx 5w), a value that was already significantly increased over normoxia (16 ± 1.7%), and it progressed to 65 ± 1.7 in the vehicle-treated group (SuHx 8w) (Fig 7A and 7B). The selexipag-treated group showed a significant decrease to 29 ± 3.0% compared with both the vehicle-treated and the pre-treatment group. Ki-67-positive cells were observed in the luminal layer or neointima of occluded small vessels, but rarely

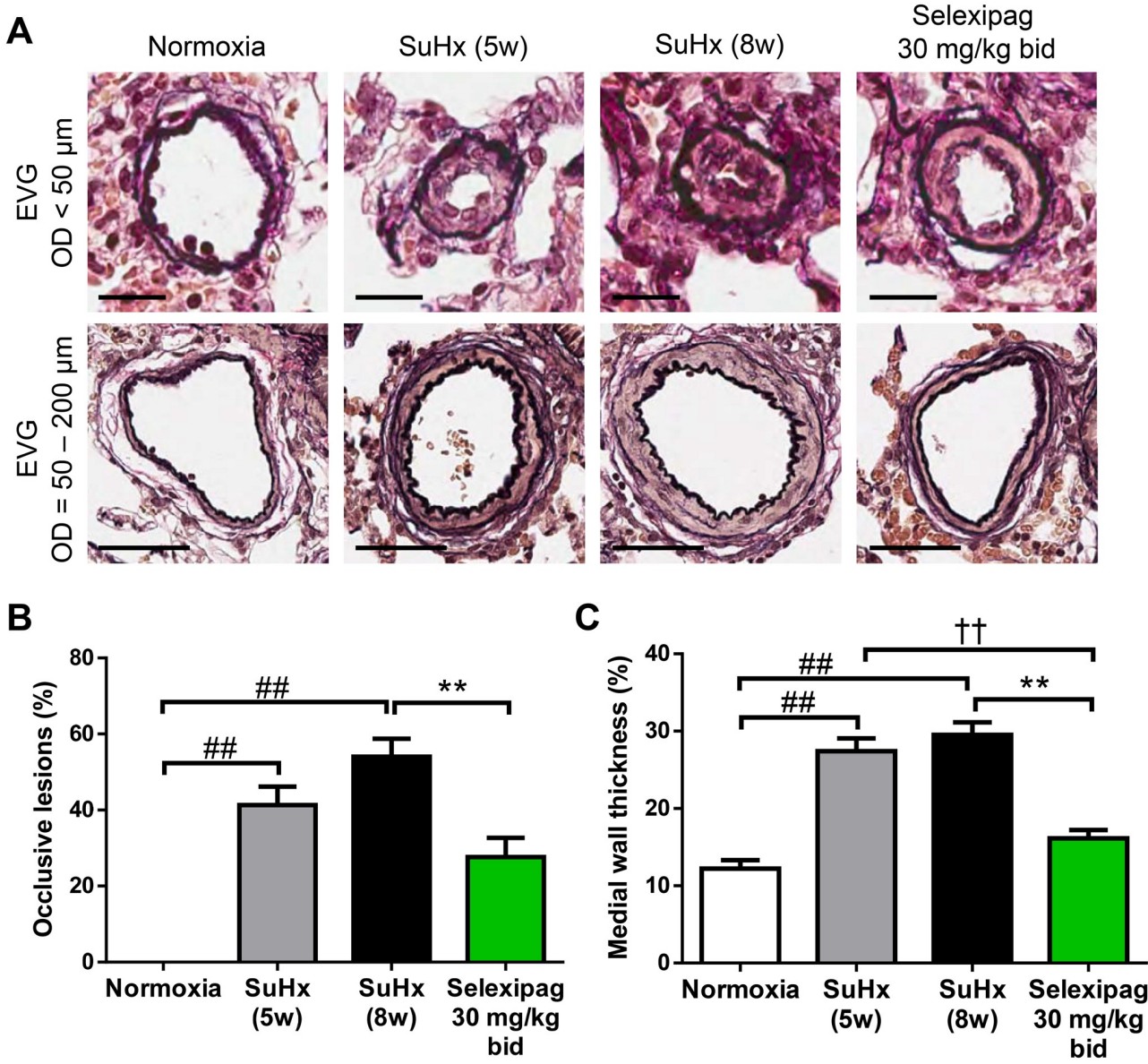

**Fig 6. Effect of selexipag on occlusive lesions in the late stage of SuHx rats.** Representative photographs of occlusive lesions (top; scale bar = 20 μm) and medial wall thickness of pulmonary vessels (bottom; scale bars = 50 μm) (A), the percentage of occlusive lesions among the vessels (B) and the percentage of medial wall thickness (C). Statistical analyses were performed using Tukey's test. ##$P$<0.01 vs. normoxia, ††$P$<0.01 vs. SuHx-5w, **$P$<0.01 vs. SuHx-8w. Values are means ± S.E.M. N = 10 per group.

in vessels of normal appearance. To investigate the lineage of the Ki-67-positive cells, we performed immunostaining with antibody against von Willebrand factor (vWF), a marker of endothelial cells, and carefully selected the same vessels which harbored Ki-67 positive cells. vWF was stained in the monolayered intima but not in the media or adventitia in the non-occluded and partially occluded vessels, whereas Ki-67-positive cells were observed in both the monolayered intima and the media in the partially and completely occluded vessels. We also assessed cell apoptosis in the occluded vessels by TUNEL staining, a method for identifying apoptotic cells by detecting DNA fragmentation. TUNEL-positive cells were more apparent in

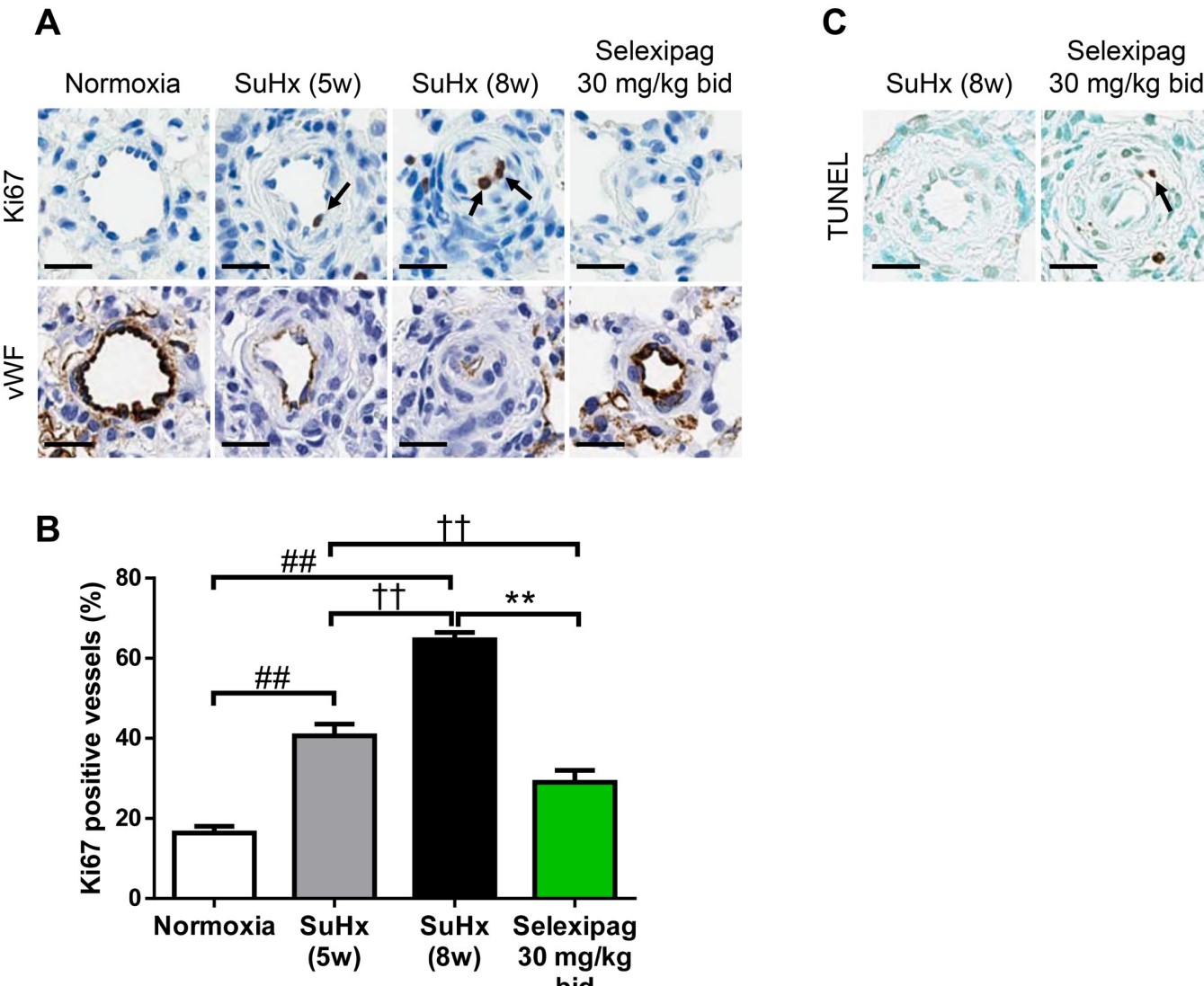

**Fig 7. Effect of selexipag on cell proliferation and apoptosis in occlusive lesions in the late stage.** Cells positive for the cell proliferation marker Ki-67 and terminal deoxynucleotidyl transferase-mediated 2'-deoxyuridine 5'-triphosphate nick-end labeling (TUNEL), a method for identifying apoptotic cells, were detected immunohistochemically (arrows). Representative photomicrographs of vessels from each group (top, Ki-67; bottom, von Willebrand factor; scale bars = 20 μm) (A), percentage of Ki-67-positive vessels (B) and representative photomicrographs of vessels with TUNEL-positive cells (C). Statistical analysis was performed using Tukey's test. ##$P<0.01$ vs. normoxia, ††$P<0.01$ vs. SuHx-5w, **$P<0.01$ vs. SuHx-8w. Values are means ± S.E.M. N = 10 per group.

the intima-media complex of the occluded lung vessels of the selexipag-treated group than in the vehicle-treated group (Fig 7C).

## Effect of selexipag on fibrosis in small lung vessels in the late stage of SuHx rats

We immunohistologically assessed the expression of collagen type I, an extracellular matrix protein associated with fibrosis, in lung vessels. As with the proportion of occluded lung vessels, the signal intensity of anti-collagen type I antibody was significantly increased in the pretreatment group (SuHx 5w; $1.06 \pm 0.04$ pixels/μm$^2$; normoxia, $0.49 \pm 0.03$ pixels/μm$^2$), with no

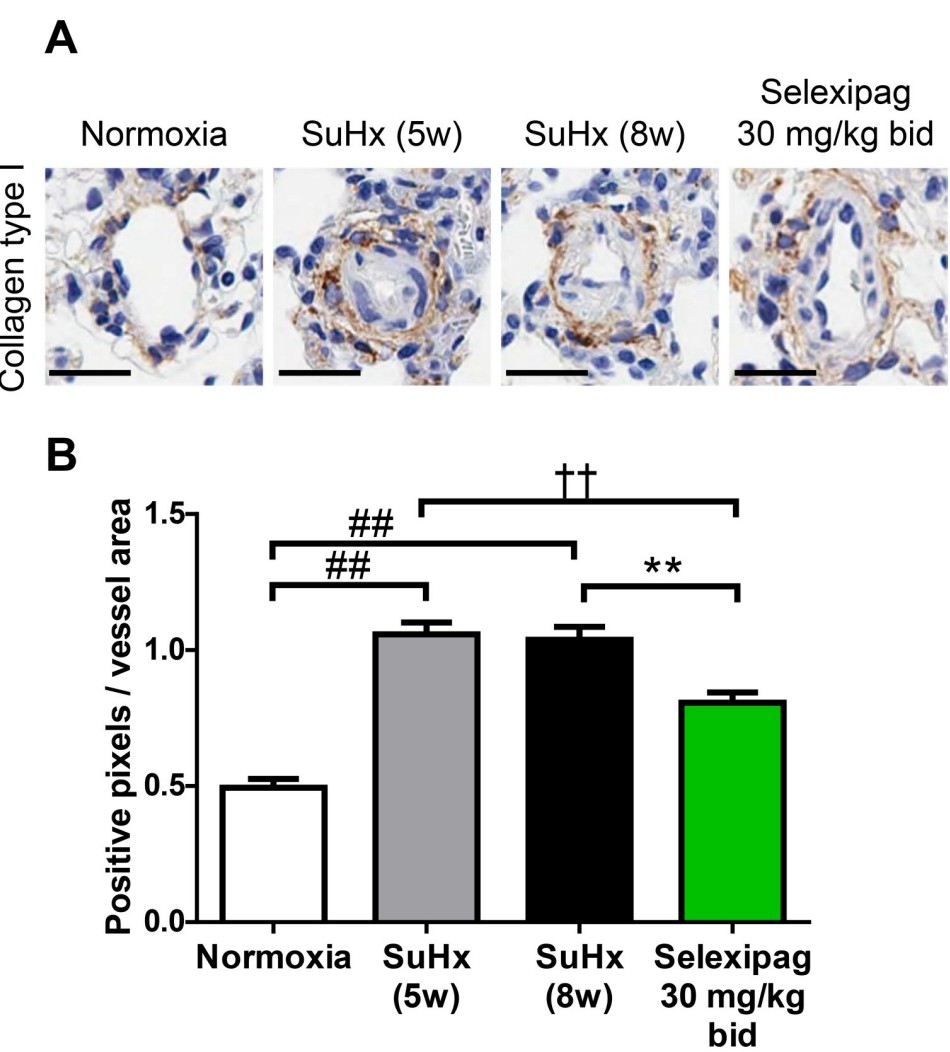

**Fig 8. Effect of selexipag on expression of collagen type I in the late stage.** Collagen type I was detected immunohistochemically. Representative photomicrographs of vessels from each group (scale bar = 20 μm) (A) and the number of positive pixels per square micrometer of vessel area (B). Statistical analysis was performed using Tukey's test. ##$P$<0.01 vs. normoxia, ††$P$<0.01 vs. SuHx-5w, **$P$<0.01 vs. SuHx-8w. Values are means ± S.E.M. N = 10 per group.

further change in the vehicle-treated group (SuHx 8w; 1.04 ± 0.05 pixels/μm$^2$) (Fig 8). The selexipag-treated group showed a significant reduction in signal intensity to 0.81 ± 0.04 pixels/μm$^2$ compared with both the vehicle-treated and the pre-treatment group.

## Effect of selexipag on mortality and RV hypertrophy in an SuHx-induced Fischer rat model of PAH

To assess the beneficial effect of selexipag on mortality by RV failure in PAH, we administered selexipag or vehicle to Fischer rats with SuHx-induced PAH for 21 days after the end of exposure to hypoxia and recorded the mortality of each group (N = 10 per group). The first death in the vehicle-treated group occurred on day 34 after Sugen 5416 injection, and the survival rate was 30% on day 42 (Fig 9A). Seven rats in the vehicle-treated group died of right heart

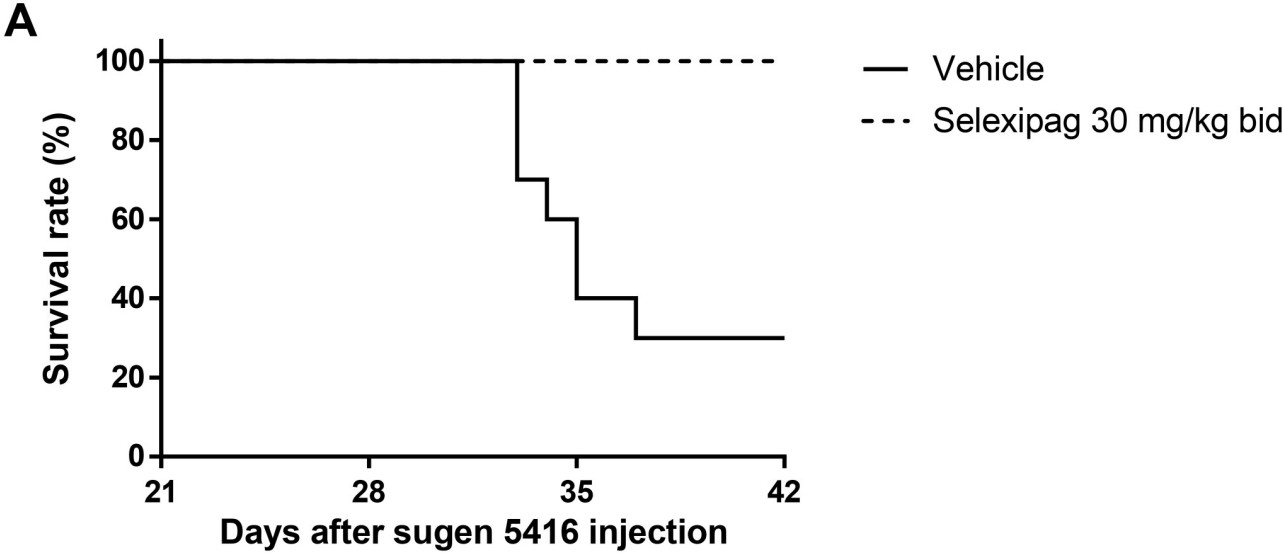

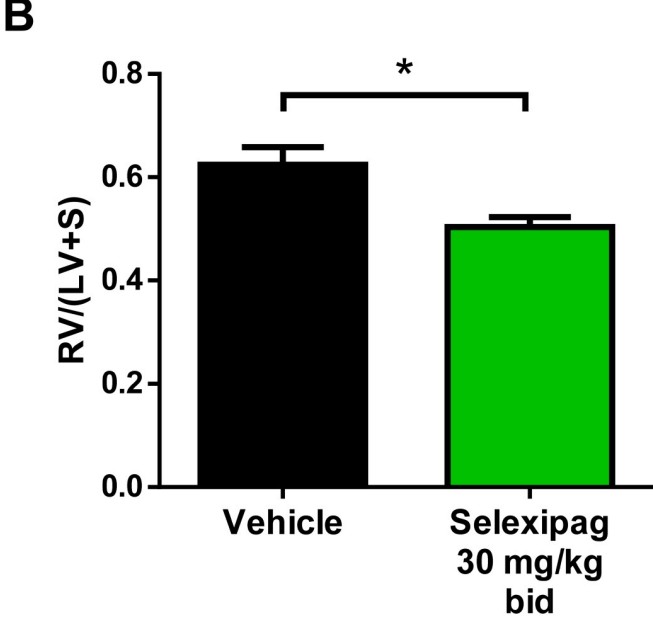

**Fig 9. Effect of selexipag on mortality and RV hypertrophy in Fischer rats with SuHx-induced PAH.** Kaplan-Meier survival curves for the two groups (A) and RV/LV+S in the surviving animals on day 42 (B). Statistical analysis was performed using log-rank analysis for survival and Student's *t*-test for RV/LV+S. *$P<0.05$ vs. vehicle. N = 10 per group.

failure. All animals in the selexipag-treated group survived until day 42, and the mortality of the selexipag-treated group was significantly lower than that of the vehicle-treated group ($P<0.01$; log-rank test). We sacrificed all surviving animals on day 42 to measure their RV hypertrophy. The RV hypertrophy was significantly improved in the selexipag-treated group compared with the vehicle-treated group (vehicle, 0.62 ± 0.03; selexipag, 0.50 ± 0.02; $P<0.05$; Student's *t*-test; Fig 9B).

## Discussion

In this study, we demonstrate the pharmacological effect of selexipag on the pathological features of PAH in the SD SuHx rat model and on mortality by RV failure in the Fischer SuHx rat model. A three-week treatment with selexipag significantly improved not only RV hypertrophy and hemodynamic parameters such as RVSP but also occlusive vascular remodeling in both the early and the late stage in SD SuHx rats. Furthermore, selexipag also attenuated the mortality rate of Fischer SuHx rats.

We have previously reported that an active metabolite of selexipag, MRE-269, has potent vasodilator effects not only on strips of lung vessels dissected from healthy humans, pigs and rats, but also on remodeled lung vessels dissected from rats with monocrotaline (MCT)-induced PAH [17,30]. This suggests that selexipag has the potential to induce vasodilation in lung vessels not only in normal rats but also in PAH model rats. However, the acute vasodilator effects of selexipag on RVSP and homeostatic systemic blood pressure have not yet been investigated in an *in vivo* model. The elevation of RVSP in a U46619-induced acute PAH model mimics the increase in vascular tone in the pathology of PAH. Our results show that selexipag effectively vasodilated the lung vessels and decreased RVSP with no effect on systemic blood pressure. Consistent with the rapid hydrolysis of selexipag to MRE-269 and the long half-life of MRE-269 in rats [31], the vasodilator effect of selexipag was stably maintained for at least 2 h, the last time point of the experiments. This long-lasting pulmonary vasodilation should have a favorable effect on the pathology of PAH, and the small effect on the systemic resistance vessels suggests a low risk of hypotension as an adverse effect.

To investigate the pharmacological effects of selexipag on the phenotypes of PAH in the SuHx model, we designed two protocols for the time course of treatment. The time course of development of the PAH phenotype in the SD SuHx model varies slightly among reports [10,32]. In our experiments, SD SuHx rats showed significant increases in RVSP and the proportion of occlusive lesions observed three weeks after Sugen 5416 injection, and both were further increased at six weeks. So we decided on treatment with selexipag from three to six weeks as the early-stage protocol to assess its suppressive effects on the progressive disease and from five to eight weeks as the late-stage protocol to assess its therapeutic effects on the established disease. Selexipag significantly improved the increases in RVSP, RV hypertrophy and occlusive vascular remodeling in both the early and the late stage. Moreover, selexipag also showed improvements compared with the pre-treatment group in the late stage. The medial wall thickness of pulmonary arteries was also attenuated by selexipag treatment in the late stage. These results suggest that selexipag has beneficial effects on not only the hemodynamics but also on occlusive vascular remodeling in established PAH pathology.

Selexipag significantly improved the hemodynamics and RV hypertrophy at doses of 20–60 mg/kg/day in SD SuHx rats and 6 mg/kg/day in MCT-induced PAH rats, whereas the approved clinical dose for patients with PAH is up to 3.2 mg/day. The area under plasma concentration–time curve from 0 to infinity ($AUC_{0-\infty}$) of MRE-269 after oral administration of 60 mg/kg/day selexipag in rats is estimated to be about 100-fold higher than the $AUC_{0-\infty}$ of MRE-269 after oral administration of 3.2 mg selexipag in healthy humans [6,33]. The difference in the dose setting of selexipag between the rat model and human patients is partly owing to the species differences in the binding affinity of selexipag for the prostacyclin receptor. Thus, the binding affinity of MRE-269 for the human prostacyclin receptor is about 10-fold higher than for the rat prostacyclin receptor [27]. In addition, like selexipag, treprostinil improves the hemodynamics of an SuHx rat model at a dose over 10-fold higher than that used in the clinical or monocrotaline-induced PAH rat model [16]. The difference in the dose required to show effectiveness may be due to differences in the severity of PAH in the models.

The other prostacyclin receptor agonists iloprost and treprostinil have been assessed for their efficacy in SD SuHx rats. Thus, inhalation of an aerosol containing iloprost for two weeks improves RV function and partially reverses RV fibrosis in SuHx rats, but does not improve RV hypertrophy or the elevation of pulmonary vascular resistance [15]. On the other hand, continuous subcutaneous infusion of treprostinil with an osmotic pump for three weeks improves both hemodynamics and RV function in SD SuHx rats, but does not improve pulmonary vascular remodeling [16]. Although an echocardiographic analysis of the effects of selexipag on RV function and structure in animal models has not yet been reported, an increase in cardiac output was observed in PAH patients treated with selexipag in a clinical study [34]. This suggests that, as with iloprost and treprostinil, an improvement in RV function may have contributed to the improved hemodynamics observed in the selexipag-treated group in our study. In contrast to iloprost and treprostinil, selexipag improved occlusive vascular remodeling in SD SuHx rats. Hemodynamic unloading by pulmonary artery banding and reducing blood flow in the lungs reverses occlusive vascular lesions in the SuHx model [35]. This suggests that a strong reduction in hemodynamic stress by pulmonary vasodilation may improve pulmonary vascular remodeling. Of all prostacyclin receptor agonists tested, selexipag shows the most potent vasodilator effect on dissected rat small pulmonary artery, probably due to its high selectivity for the prostacyclin receptor and low binding affinity for other vasoconstrictive prostaglandin receptors [17,27]. Furthermore, selexipag does not induce activation of the β-arrestin pathway or desensitization of the prostacyclin receptor, and the vasodilator effect of selexipag on skin blood flow in rats is not attenuated by repeated administration [27,36]. It is probably because of these pharmacological features that selexipag decreased hemodynamic stress in our SuHx rats more effectively than did other prostacyclin receptor agonists.

Several groups have reported that alpha-smooth muscle actin (SMA)-positive smooth muscle cells are observed in both the neointima and the media in occlusive pulmonary vessels in SD SuHx rats. On the other hand, vWF-positive endothelial cells are observed in the monolayered intima, but not in the media [37–39]. In the selexipag-treated group, we observed Ki-67-positive cells not only in the neointima and media which occluded the small vessels but also in the monolayered intima which showed vWF staining. This suggests that the proliferation of both smooth-muscle-like cells in the neointima and media and endothelial-like cells in the monolayered intima were activated in the lung vessels of SuHx rats. It is in good agreement with the pathology of PAH patients that the abnormalities of both smooth muscle cells and endothelial cells are important in the development of vascular remodeling [40]. In the present study, consistent with the reduction in vascular occlusive lesions, the numbers of Ki-67-positive cells were decreased and TUNEL-positive apoptotic cells were observed in the intima-media complex in the occluded lung vessels in the selexipag-treated group. It seems likely that TUNEL staining was shown by smooth-muscle-like cells in both the neointima and the media. The antiproliferative or proapoptotic effect of selexipag on vascular smooth muscle cells or endothelial cells dissociated from SD SuHx rats has not been tested. However, prostacyclin receptor agonists, including selexipag, are known to have an antiproliferative effect on human pulmonary arterial smooth muscle cells stimulated by platelet-derived growth factor [36], whereas direct effects of selexipag on vascular endothelial cells and apoptosis have not yet been reported. The antiproliferative effect of selexipag on vascular smooth muscle cells may also contribute to the improvement of occlusive vascular remodeling. The detailed mechanism of the anti-vascular remodeling effects of selexipag needs further investigation.

Selexipag decreased collagen type I expression in the adventitia of lung vessels. In addition, in normal human lung fibroblasts, MRE-269 significantly inhibited the transforming growth factor (TGF) β1-stimulated production of procollagen type I C-peptide (S1 Fig). Another

group has also reported that selexipag has antifibrotic effects on human fibroblasts stimulated by platelet-derived growth factor [41]. In addition to the potent pulmonary vasodilator effect of selexipag, its direct effects on vascular smooth muscle cells and fibroblasts may have contributed to its improvement of pulmonary vascular remodeling and fibrosis in SuHx rats.

Selexipag attenuated the development of RV hypertrophy and reduced the mortality associated with RV failure in SuHx rats. As far as we know, this is the first demonstration of a reduction in mortality risk in an SuHx rat model among drugs approved for PAH treatment. Increased afterload due to persistent high pulmonary arterial pressure causes RV hypertrophy and eventually leads to RV failure in patients with PAH [42,43]. The reduction in mortality obtained in our study probably resulted from a decrease in the afterload of the RV through pulmonary vasodilation with a resulting improvement in pulmonary vascular remodeling.

In conclusion, selexipag improved hemodynamics, occlusive vascular remodeling and mortality in SuHx-induced rat models of severe PAH. These effects were probably based on the potent prostacyclin receptor agonistic effect of selexipag on pulmonary vessels. This is the first report to show that activation of the prostacyclin receptor exerts therapeutic effects on not only the hemodynamics but also on occlusive vascular remodeling and mortality by RV failure in SuHx rat models. Selexipag has been approved for and is in use in the clinical treatment of PAH worldwide. It is likely that these beneficial effects on multiple aspects of PAH pathology contribute to the clinical outcomes in patients with PAH.

## Supporting information

**S1 File. Supplemental materials and methods.**
(DOCX)

**S1 Fig. Effect of MRE-269 on TGFβ-induced collagen production of normal human lung fibroblasts.** Statistical analyses were performed using Student's t- test followed by Dunnett's test. ##$P<0.01$ vs. TGFβ (-) group by Student's t-test, **$P<0.01$ vs. TGFβ (+) group by Dunnett's test. Values are means ± S.E.M. N = 4 per group.
(PPTX)

## Acknowledgments

We thank Dr. Gerald E. Smyth for helpful suggestions during the preparation of the manuscript and Dr. Michiko Oka for useful advice on experiments and documentation.

## Author Contributions

**Conceptualization:** Keiji Kosugi, Keiichi Kuwano.

**Data curation:** Yohei Honda.

**Formal analysis:** Yohei Honda, Chiaki Fuchikami, Kazuya Kuramoto, Yuki Numakura.

**Investigation:** Yohei Honda, Keiji Kosugi, Chiaki Fuchikami.

**Methodology:** Keiji Kosugi.

**Project administration:** Keiichi Kuwano.

**Supervision:** Keiichi Kuwano.

**Validation:** Kazuya Kuramoto.

**Visualization:** Yohei Honda, Kazuya Kuramoto.

**Writing – original draft:** Yohei Honda, Keiji Kosugi, Kazuya Kuramoto.

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
