## [Decision Letter · Decision Letter 0]

31 Jul 2020

PONE-D-20-17665

The selective PGI2 receptor agonist selexipag ameliorates Sugen 5416/hypoxia-induced pulmonary arterial hypertension in rats

PLOS ONE

Dear Dr. Honda,

Thank you for submitting your manuscript to PLOS ONE. After careful consideration, we feel that it has merit but does not fully meet PLOS ONE’s publication criteria as it currently stands. Therefore, we invite you to submit a revised version of the manuscript that addresses the points raised during the review process.

We look forward to receiving your revised manuscript.

Kind regards,

Michael Bader

Academic Editor

PLOS ONE

Journal Requirements:

2.At this time, we request that you  please report additional details in your Methods section regarding animal care, as per our editorial guidelines:

(1) Please state the number of rats used in the study  

(2) Please describe the post-operative care received by the animals, including the frequency of monitoring and the specific clinical, physiological and behavioural criteria used to assess animal health and well-being.

Thank you for your attention to these requests.

3.Thank you for stating the following financial disclosure:

 [All authors are employed by Nippon Shinyaku Co., Ltd.].               

4.Thank you for stating the following in the Competing Interests section:

[All authors are employed by Nippon Shinyaku Co., Ltd.].

Reviewers' comments:

Reviewer's Responses to Questions

**Comments to the Author**

1. Is the manuscript technically sound, and do the data support the conclusions?

Reviewer #1: Yes

Reviewer #2: Yes

2. Has the statistical analysis been performed appropriately and rigorously? 

Reviewer #1: Yes

Reviewer #2: Yes

3. Have the authors made all data underlying the findings in their manuscript fully available?

Reviewer #1: Yes

Reviewer #2: Yes

4. Is the manuscript presented in an intelligible fashion and written in standard English?

Reviewer #1: Yes

Reviewer #2: Yes

5. Review Comments to the Author

Reviewer #1: Comments to the authors:

Honda and colleagues have conducted a study of the effects of selexipag on rat models of pulmonary arterial hypertension. Authors used three animal models (the acute dose of U46619 and Sugen/Hypoxia on two rat strains, Sprague Dawley and Fischer). Honda et al. show the ability of selexipag to attenuate the vasoconstrictive effect of U46619. They also showed the decrease in RVSP and Fulton’s index in the SuHx Sprague Dawley rats (2 time-points), as well as the decreased mortality in Fischer rats. In the SuHx model Honda et al showed a decrease in KI-67 positive cells with an increase in apoptotic cells and a decrease in collagen deposition in the lung vasculature.

Major points:

1. The authors decided to use term developing-phase and established-phase of PAH when referring to the morphological changes in the lung, however the readership may interpret it as a development of PAH itself, which is not correct: In the previous, comprehensive study of the SuHx rat PAH model, Legchenko et al, (Sci Trans Med, 2018) show that at the 3+1 week (1 week post-weeks hypoxia hypoxia) animals do have an established PAH phenotype, and in the following 5 weeks right heart failure develops. Thus, I would recommend to call the stages the authors did their experiments in as “early” and “late-stage”.

2. While selexipag indeed significantly decreased the RVSP in SuHx rats, the pressures still remained suprasytemic (from 141 to 102 mmHg). This needs to be mentioned since basically the rats still have severe PAH: Most other groups find average RVSP of 90-110mmHg, and some groups found average RVSP of only 50-70mmmHg in SuHx rats. Bogaard HJ et al. AJRCCM, 2019, doi: 10.1164/rccm.201906-1200LE.

2. For the TUNEL assay it would be beneficial to have the aSMA staining (on the next serial section slide) to show the apoptotic cells are smooth muscle cells and not endothelial.

3. In the conclusion: lines 585-586: “…taken together with those obtained for SuHx rats in the present study, suggest that selexipag would have beneficial effects on multiple aspects of PAH pathology”.

The authors need to re-write these conclusions as selexipag has been approved (as authors write themselves in the introduction, line 69-70) and is widely used in clinical care for > 3 years.

5. Many cited references are outdated.

Critique by section

ABSTRACT:

Wording could be improved. Write Sprague Dawley strain and Fischer strain – so far only the Fischer strain is mentioned at the end of the abstract.

INTRODUCTION:

Exchange references, as listed below under REFERENCES.

METHODS/DISCUSSION:

The authors should discuss the straindependency of the phenotype of the SuHx rat model, with best results in the Sprague Dawley rats who lack a emphysema phenotype (whie WKY rats were reported by one group to have emphysema disqualifying the WKY rat SuHx as model for PAH):

References to be cited ad discussed:

The Adult Sprague-Dawley Sugen-Hypoxia Rat Is Still "the One:" A Model of Group 1 Pulmonary Hypertension: Reply to Le Cras and Abman.

Bogaard HJ, Legchenko E, Ackermann M, Kühnel MP, Jonigk DD, Chaudhary KR, Sun X, Stewart DJ, Hansmann G.

Emphysema Is-at the Most-Only a Mild Phenotype in the Sugen/Hypoxia Rat Model of Pulmonary Arterial Hypertension.

Bogaard HJ, Legchenko E, Chaudhary KR, Sun XQ, Stewart DJ, Hansmann G. Am J Respir Crit Care Med. 2019 Dec 1;200(11):1447-1450. doi: 10.1164/rccm.201906-1200LE.

Rational for choosing SuHx: The SuHx rat is considered superior vs. MCT and other rodent models by a group of experts: Bonnet S et al. Translating Research into Improved Patient Care in Pulmonary Arterial Hypertension, Am J Resp Crit Care Med, 2017. Also, while near every intrervention seems to work in the monocrotain /MCT) rat PAH model, this is clearly not the case in SuHx exposed rats that are rather treatment resistant.

The most comprehensive phenotyping study on the SUHx rat model, by means of echo, right and left heart cath, MRI and PET CT has been conducted by Legchenko et al. and this paper should be cited and discussed (see experimental design, RV assessment, PVD assessment):

PPARγ agonist pioglitazone reverses pulmonary hypertension and prevents right heart failure via fatty acid oxidation.

Legchenko E, Chouvarine P, Borchert P, Fernandez-Gonzalez A, Snay E, Meier M, Maegel L, Mitsialis SA, Rog-Zielinska EA, Kourembanas S, Jonigk D, Hansmann G. Sci Transl Med. 2018 Apr 25;10(438):eaao0303. doi: 10.1126/scitranslmed.aao0303.

Why did the authors use male Sprague Dawley rats (original ad standard strain) for most experiments but Fischer rats (male ?) for the survival experiment ?

Just let us know the rationale of change of protocol ?

What was the mean RVSP that the authors measured in Fischer rats (in SD, the authors report 130-140mmHg !! which is international record for the average RVSP in this SuHx model)

As reported by authors, Fischer and Sprague-Dawley rats developed similar increases in RVSP to 100 mm Hg.

See: Jiang B, Deng Y, Suen C, Taha M, Chaudhary KR, Courtman DW, et al.

Marked strain-specific differences in the SU5416 rat model of severe

pulmonary arterial hypertension. Am J Respir Cell Mol Biol 2016;54:

461–468.

It seems Selexipag was given s.c. twice daily subcutaneously, please write this clearly

So far: “Selexipag was suspended in 0.5% methylcellulose solution and 10 or 30 mg/kg of the compound was administered twice daily.”

This mmeas the total daily dosse of Selexipag was 20-60mg/rat/day while the current max. recommended adult patient dose is 1.6 mg (1600 mcg) twice daily PO. Please discuss in the discussion why such a super physiological dose was used.

Why did the authors not incorporate the drug into the food, especially in the reversal experiments, as described by Legchenko et al. Science Translational Medicine, 2018 ?

Both Sugen (SU5416) and selexipag was dissolved in 0.5% methylcellulose solution … to acieve a complete suspension or even clear solutiio with 0.5% methylcellulose is nearly impossible, and probably responsibe for the great variattey of RVSP reported and variance within the same study. DMSO as vehicle ad solvent would have been an alternative. Was the controls the authors used a vehicle control (0.5% methylcellulose), meaning where the rats not treated with selelcipag s.c. ttwice daily injected twiche daily with 0.5% methylcellulose vol../vol. ? Twice daily injections are quite stressful,a ddn if the authors have not done the control injections, this must be listed under limiitations of the study.

Systemic blood pressure was measured by tail cuff, which inferior to invasive measure of aortic and LV enddiastolic pressure. However, invasive SAP and LVEDDP was not different between the groups, as shown by Legchhenko et al. (no postcapillary PH). This observation should be cited, but then I think it is ok to have only the tail cuff data.

RESULTS:

Figure 1. What were the absolute changes in mmHg ? Pllease write that at least in the text. It is hard to crasp from the figure that showns only percentage changes.

Figure 1B: are the two last data points indeed not significant?

Figure 2. indicates that al rats received vehicle s.c. – good.

Figure 3. I suggest to use color coding for the different groups in stead of multipe difrenet b/w patterns. Same for the other figures with multiple columns.

In the authors’ ATS abstract from 2017, the RVSP in SuHx rats with PA was very high (ca- 130-140 mmHg), and could be significantly decreased to 100 mmHg, but only with the higher Selxipag dose of 30mg/kg/dose s.c. twice daily.

https://www.atsjournals.org/doi/abs/10.1164/ajrccm-conference.2017.195.1_MeetingAbstracts.A4221

The Novel and Selective Prostacyclin Receptor Agonist Selexipag Vasodilates Human Pulmonary Arteries in an Endothelium-Independent Manner and Ameliorates Pulmonary Arterial Hypertension in Rat Models

Kazuya Kuramoto , Chiaki Fuchikami , Keiji Kosugi , Yohei Honda , Michiko Oka , . American Journal of Respiratory and Critical Care Medicine 2017;195:A4221

Please make sure that all authors of this ATS abstract are also listed as authors on this original article

Kazuya Kuramoto , Chiaki Fuchikami , Keiji Kosugi , Yohei Honda , Michiko Oka , Keiichi Kuwano

i.e., should Michiko Oka not be an author on the paper ? If not, please provide the freasonig why this has been changed after an abstract with the name of this author as been published.

Figure 3. What is known about the effect of selexipag on RV systolic and diastolic function in SuHx exposed rats ? The authors should provide the (blinded) RVEDP measurements, RV dp/dt max and min. It would be greta if the authors had echo data but it seems it would require the entire study if they do not have those data.

Figure 4A, 6A, 8A – include the name of the staining in the figure (as in Figure 7).

Figure 6B – 1st column can’t be empty as you performed statistics on it.

Figure 7. As far as I know, it was not know that selecipag ca induce apoptosis in the media (TUNEL assay), probably SMC. Can this not simpy explained by the supraphysiological doses ? Can the authors provide cell culture experimental data that underpins this finding with much lower selexipag’s active metabolite in culture, in the dose range of 10nM – 1uM ?

Overall, I feel the figures could be also in panels so that the total number of figures can be reduced to approx.. 7, but leave this to the discretion of the handling editor.

Have the authors performed any expression studies on the RV and LV, or even only whole lung, to get a handle on the RNA/protein changes induced by selexipag and its active metabolite…. ?

Have the authors performed any expression studies on other organs, such as liver and idney that are probably also affected by the demonstrated systemic to suprasystemic RV and PA pressures in SuHx rats ?

Minor points:

1. In the introduction after approval (lines 69-70), it is important to address the use of selexigag in pediatric patients (Geerdink et al, Pulm Circ 2017; Koo et al., Cardiol Young, 2019; Rothman et al, Pulm Circ 2020; Hansmann et al, J H Lung Transplant 2020).

2. There is data not included in the figues in the discussion (lines 532-536). Please, include in the figures/supplemental figures.

3. Discussion line 568: “Selexipag partially reversed RV hypertrophy” – you haven’t shown a direct effect of selexipag on the heart or isolated cardiomyocytes, also as was shown in Legchenko et al.,Sci Trans Med, 2018, at 3+1 weeks, RV diameter is not changed in Sprague Dawley rats, so at the time of administration of selexipag, the RV was not hypertrophied. Needs to be re-written as: administration of selexipag attenuated the development of RV hypertrophy.

Reviewer #2: Honda et al. investigated the effects of selexipag in sugen/hypoxia induced PAH. Selexipag has acute vasodilator effect and improved RVSP, RVH and pulmonary artery remodeling in prevention and treatment protocol. This study is interested; however, some concerns are included.

1. What caused the reverse pulmonary artery remodeling? Could reduction of shear stress by vasodilation of selexipag reverse pulmonary artery remodeling? Or Did selexipag directly affect pulmonary arteries?

2. Authors should emphasize the novelty of this study. Similar studies had already published.

3. Did authors examine the proapoptotic and antiproliferative effects in pulmonary artery endothelial or smooth muscle cells from sugen/hypoxia rats?

4. How many dose of selexipag 30mg/kg is in human setting?

6. PLOS authors have the option to publish the peer review history of their article (what does this mean?). If published, this will include your full peer review and any attached files.

Reviewer #1: No

Reviewer #2: No

---

## [Author Response · Author response to Decision Letter 0]

2 Sep 2020

Prof. Dr. Michael Bader

Academic Editor

PLOS ONE

Dear Dr. Bader:

PLOS ONE PONE-D-20-17665

The selective PGI2 receptor agonist selexipag ameliorates Sugen 5416/hypoxia-induced pulmonary arterial hypertension in rats

Thank you very much for your e-mail of August 1st 2020, and for the review of our manuscript.

I hope that you’ll find my response to the comments satisfactory. I am re-submitting a revised manuscript for consideration of publication in PLOS ONE.

 Thank you very much for your consideration.

 Sincerely yours,

Yohei Honda

Enclosures:

 

Journal Requirements:

(Ans.) We have ensured that our manuscript meets PLOS ONE's style requirements.

2.At this time, we request that you please report additional details in your Methods section regarding animal care, as per our editorial guidelines:

(1) Please state the number of rats used in the study 

(2) Please describe the post-operative care received by the animals, including the frequency of monitoring and the specific clinical, physiological and behavioural criteria used to assess animal health and well-being.

(Ans.) We have rewritten the appropriate parts of the Methods section according to your suggestion.

3.Thank you for stating the following financial disclosure:

 [All authors are employed by Nippon Shinyaku Co., Ltd.]. 

(Ans.) We have already stated it in the cover letter. 

4.Thank you for stating the following in the Competing Interests section:

[All authors are employed by Nippon Shinyaku Co., Ltd.].

(Ans.) We have already stated it in the cover letter.

 

Reviewers' comments:

We deeply appreciate your kind advice on our manuscript. Bearing your comments in mind, we have revised the manuscript to make our conclusions more convincing. Below we reply to the specific points you raised.

Reviewer #1: Comments to the authors:

Honda and colleagues have conducted a study of the effects of selexipag on rat models of pulmonary arterial hypertension. Authors used three animal models (the acute dose of U46619 and Sugen/Hypoxia on two rat strains, Sprague Dawley and Fischer). Honda et al. show the ability of selexipag to attenuate the vasoconstrictive effect of U46619. They also showed the decrease in RVSP and Fulton’s index in the SuHx Sprague Dawley rats (2 time-points), as well as the decreased mortality in Fischer rats. In the SuHx model Honda et al showed a decrease in KI-67 positive cells with an increase in apoptotic cells and a decrease in collagen deposition in the lung vasculature.

Major points:

1. The authors decided to use term developing-phase and established-phase of PAH when referring to the morphological changes in the lung, however the readership may interpret it as a development of PAH itself, which is not correct: In the previous, comprehensive study of the SuHx rat PAH model, Legchenko et al, (Sci Trans Med, 2018) show that at the 3+1 week (1 week post-weeks hypoxia hypoxia) animals do have an established PAH phenotype, and in the following 5 weeks right heart failure develops. Thus, I would recommend to call the stages the authors did their experiments in as “early” and “late-stage”. 

 (Ans.) According your suggestion, we changed the description of the experimental stages to “early” and “late-stage”. 

2. While selexipag indeed significantly decreased the RVSP in SuHx rats, the pressures still remained suprasytemic (from 141 to 102 mmHg). This needs to be mentioned since basically the rats still have severe PAH: Most other groups find average RVSP of 90-110mmHg, and some groups found average RVSP of only 50-70mmmHg in SuHx rats. Bogaard HJ et al. AJRCCM, 2019, doi: 10.1164/rccm.201906-1200LE.

(Ans.) In the revised manuscript, we state that rats in the selexipag-treated group had still had high RVSP. We had calibrated the measurement system with a mercury manometer before each experiment and this is now mentioned in the Methods section (lines 201-203).

2. For the TUNEL assay it would be beneficial to have the aSMA staining (on the next serial section slide) to show the apoptotic cells are smooth muscle cells and not endothelial.

(Ans.) We did not perform immunohistochemical staining with anti-alpha-SMA antibody. However, several groups have reported that, like the pathology of PAH patients, alpha-SMA-positive smooth muscle cells are observed in both neointima and media in occlusive pulmonary vessels in SD SuHx rats. On the other hand, vWF-positive endothelial cells are observed in monolayered intima, but not in media [Tamura et al., 2018, Jernigan et al., 2017, Mair et al., 2014]. In our study, TUNEL-positive cells were observed in both neointima and media; this suggests that at least some of the TUNEL-positive cells would be smooth muscle cells. We have added a description of these previous reports and references to the discussion section (lines 581-585).

Tamura Y, Phan C, Tu L, Hiress M, Thuillet R, Jutant EM et al.

Ectopic upregulation of membrane-bound IL6R drives vascular remodeling in pulmonary arterial hypertension

Clin Invest. 2018;128(5):1956-1970. doi: 10.1172/JCI96462

Mair K, Wright A, Duggan N, Rowlands D, Hussey M, Roberts S et al.

Sex-dependent influence of endogenous estrogen in pulmonary hypertension

Am J Respir Crit Care Med. 2014;190(4):456-67. doi: 10.1164/rccm.201403-0483OC

Jernigan N, Naik J, Weise-Cross L, Detweiler N, Herbert L, Yellowhair T, Resta T

Contribution of reactive oxygen species to the pathogenesis of pulmonary arterial hypertension

PLoS One. 2017;12(6):e0180455. doi: 10.1371/journal.pone.0180455

3. In the conclusion: lines 585-586: “…taken together with those obtained for SuHx rats in the present study, suggest that selexipag would have beneficial effects on multiple aspects of PAH pathology”.

The authors need to re-write these conclusions as selexipag has been approved (as authors write themselves in the introduction, line 69-70) and is widely used in clinical care for > 3 years.

(Ans.) According your suggestion, we mentioned that selexipag has been approved in Discussion and Abstract section (lines 632-633 and lines 51-52).

5. Many cited references are outdated.

(Ans.) According your suggestion, we updated references.

Critique by section

＊ABSTRACT:

Wording could be improved. Write Sprague Dawley strain and Fischer strain – so far only the Fischer strain is mentioned at the end of the abstract.

(Ans.) We have improved the wording as you suggest (line 40).

＊INTRODUCTION:

Exchange references, as listed below under REFERENCES.

(Ans.) According your suggestion, we updated references.

＊METHODS/DISCUSSION:

The authors should discuss the straindependency of the phenotype of the SuHx rat model, with best results in the Sprague Dawley rats who lack a emphysema phenotype (whie WKY rats were reported by one group to have emphysema disqualifying the WKY rat SuHx as model for PAH):

References to be cited ad discussed:

The Adult Sprague-Dawley Sugen-Hypoxia Rat Is Still "the One:" A Model of Group 1 Pulmonary Hypertension: Reply to Le Cras and Abman.

Bogaard HJ, Legchenko E, Ackermann M, Kühnel MP, Jonigk DD, Chaudhary KR, Sun X, Stewart DJ, Hansmann G.

Emphysema Is-at the Most-Only a Mild Phenotype in the Sugen/Hypoxia Rat Model of Pulmonary Arterial Hypertension.

Bogaard HJ, Legchenko E, Chaudhary KR, Sun XQ, Stewart DJ, Hansmann G. Am J Respir Crit Care Med. 2019 Dec 1;200(11):1447-1450. doi: 10.1164/rccm.201906-1200LE.

Rational for choosing SuHx: The SuHx rat is considered superior vs. MCT and other rodent models by a group of experts: Bonnet S et al. Translating Research into Improved Patient Care in Pulmonary Arterial Hypertension, Am J Resp Crit Care Med, 2017. Also, while near every intrervention seems to work in the monocrotain /MCT) rat PAH model, this is clearly not the case in SuHx exposed rats that are rather treatment resistant.

The most comprehensive phenotyping study on the SUHx rat model, by means of echo, right and left heart cath, MRI and PET CT has been conducted by Legchenko et al. and this paper should be cited and discussed (see experimental design, RV assessment, PVD assessment):

PPARγ agonist pioglitazone reverses pulmonary hypertension and prevents right heart failure via fatty acid oxidation.

Legchenko E, Chouvarine P, Borchert P, Fernandez-Gonzalez A, Snay E, Meier M, Maegel L, Mitsialis SA, Rog-Zielinska EA, Kourembanas S, Jonigk D, Hansmann G. Sci Transl Med. 2018 Apr 25;10(438):eaao0303. doi: 10.1126/scitranslmed.aao0303.

(Ans.) We have added a description of SuHx rat models and references you kindly mentioned to the introduction section (lines 82-98).

＊Why did the authors use male Sprague Dawley rats (original ad standard strain) for most experiments but Fischer rats (male ?) for the survival experiment ?

＊Just let us know the rationale of change of protocol ? 

 (Ans.) To date, most studies of the SuHx model have used Sprague-Dawley rats, so in most experiments we used Sprague-Dawley rats. They show a severe PAH phenotype, but show excellent survival for up to 14 weeks. To evaluate the effect of selexipag on mortality, we used Fischer rats, which exhibit very high mortality because of strain-dependent differences [Suen CM et al., 2019]. We added this point in the introduction section (lines 82-98).

Suen CM, Chaudhary KR, Deng Y, Jiang B, Stewart DJ.

Fischer rats exhibit maladaptive structural and molecular right ventricular remodelling in severe pulmonary hypertension: A genetically prone model for right heart failure. 

Cardiovasc Res. 2019;115: 788–799. doi:10.1093/cvr/cvy258

What was the mean RVSP that the authors measured in Fischer rats (in SD, the authors report 130-140mmHg !! which is international record for the average RVSP in this SuHx model)

As reported by authors, Fischer and Sprague-Dawley rats developed similar increases in RVSP to 100 mm Hg.

See: Jiang B, Deng Y, Suen C, Taha M, Chaudhary KR, Courtman DW, et al.

Marked strain-specific differences in the SU5416 rat model of severe

pulmonary arterial hypertension. Am J Respir Cell Mol Biol 2016;54:

461–468.

(Ans.) Because we have not measured RVSP in the survival study with Fischer SuHx rats, we do not have any data to compare with the RVSP of SD SuHx rats. On the other hand, the development of RV hypertrophy of Fischer SuHx rats (RV/(LV+S); 0.62 ± 0.03, Fig. 9B) was quite similar to that of SD SuHx rats (RV/(LV+S); 0.63 ± 0.02, Fig. 3B). This is in good agreement with the reference you mention, which shows that Fischer SuHx rats develop RV hypertrophy and increased RVSP at the same level as SD SuHx rats. Therefore, we guess that the RVSP of our Fischer SuHx rats would probably be increased to levels similar to those of SD SuHx rats.

＊It seems Selexipag was given s.c. twice daily subcutaneously, please write this clearly

So far: “Selexipag was suspended in 0.5% methylcellulose solution and 10 or 30 mg/kg of the compound was administered twice daily.”

(Ans.) Selexipag was given orally twice daily. This is now clearly stated in the revised Materials and Methods (lines 168-171). 

＊This mmeas the total daily dosse of Selexipag was 20-60mg/rat/day while the current max. recommended adult patient dose is 1.6 mg (1600 mcg) twice daily PO. Please discuss in the discussion why such a super physiological dose was used.

(Ans.) We have added an explanation of this point in an extra paragraph in the Discussion (lines 537-552). 

＊Why did the authors not incorporate the drug into the food, especially in the reversal experiments, as described by Legchenko et al. Science Translational Medicine, 2018 ?

(Ans.) Mixing selexipag into the food affected the amount of food intake of the SD SuHx rats. In order to ensure that every rat was given the same dose of selexipag, we administered selexipag to rats orally. 

＊Both Sugen (SU5416) and selexipag was dissolved in 0.5% methylcellulose solution … to acieve a complete suspension or even clear solutiio with 0.5% methylcellulose is nearly impossible, and probably responsibe for the great variattey of RVSP reported and variance within the same study. DMSO as vehicle ad solvent would have been an alternative. Was the controls the authors used a vehicle control (0.5% methylcellulose), meaning where the rats not treated with selelcipag s.c. ttwice daily injected twiche daily with 0.5% methylcellulose vol../vol. ? Twice daily injections are quite stressful,a ddn if the authors have not done the control injections, this must be listed under limiitations of the study.

(Ans.) Sugen was suspended in buffer containing 0.5% carboxymethyl cellulose sodium salt, 0.9% NaCl, 0.4% polysorbate 80 and 0.9% benzyl alcohol, and a single injection was given subcutaneously to induce PAH. On the other hand, selexipag was dissolved in 0.5% methylcellulose solution, and administered orally twice daily. Vehicle control rats were administered 0.5% methylcellulose solution orally twice daily instead of selexipag solution. We revised the materials and methods section to clarify the route of administration of these materials (lines 160-171). 

＊Systemic blood pressure was measured by tail cuff, which inferior to invasive measure of aortic and LV enddiastolic pressure. However, invasive SAP and LVEDDP was not different between the groups, as shown by Legchhenko et al. (no postcapillary PH). This observation should be cited, but then I think it is ok to have only the tail cuff data.

(Ans) In the U46619-induced acute PAH rat model, we measured mean arterial pressure (MAP) by inserting a cannula into the right femoral artery. This is now clearly stated in the methods section (lines 138-139). The MAP measured invasively by cannula and noninvasively by tail cuff were not very different in our experiments. So we chose the noninvasive tail cuff method to measure MAP in later experiments.

RESULTS:

＊Figure 1. What were the absolute changes in mmHg ? Pllease write that at least in the text. It is hard to crasp from the figure that showns only percentage changes.

(Ans.) We have added the absolute changes in mmHg to the Results (line 300).

＊Figure 1B: are the two last data points indeed not significant?

(Ans.) According to the statistical analysis, there was no significant difference between the two groups in mean arterial pressure.

＊Figure 2. indicates that al rats received vehicle s.c. – good.

(Ans.) Vehicle control rats received 0.5% methylcellulose solution orally. We added this point in an extra paragraph in the Materials and Methods (lines 168-169).

＊Figure 3. I suggest to use color coding for the different groups in stead of multipe difrenet b/w patterns. Same for the other figures with multiple columns.

(Ans.) We changed to color coding of figures.

＊In the authors’ ATS abstract from 2017, the RVSP in SuHx rats with PA was very high (ca- 130-140 mmHg), and could be significantly decreased to 100 mmHg, but only with the higher Selxipag dose of 30mg/kg/dose s.c. twice daily.

https://www.atsjournals.org/doi/abs/10.1164/ajrccm-conference.2017.195.1_MeetingAbstracts.A4221

The Novel and Selective Prostacyclin Receptor Agonist Selexipag Vasodilates Human Pulmonary Arteries in an Endothelium-Independent Manner and Ameliorates Pulmonary Arterial Hypertension in Rat Models

Kazuya Kuramoto , Chiaki Fuchikami , Keiji Kosugi , Yohei Honda , Michiko Oka , . American Journal of Respiratory and Critical Care Medicine 2017;195:A4221

Please make sure that all authors of this ATS abstract are also listed as authors on this original article

Kazuya Kuramoto , Chiaki Fuchikami , Keiji Kosugi , Yohei Honda , Michiko Oka , Keiichi Kuwano

i.e., should Michiko Oka not be an author on the paper ? If not, please provide the freasonig why this has been changed after an abstract with the name of this author as been published.

(Ans.) The figure of RVSP and RV hypertrophy in the ATS abstract A4221 is exactly same as Fig. 3A and 3B in this article. The administration of 30 mg/kg selexipag p.o. twice daily significantly decreased RVSP. Michiko Oka contributed to the ex vivo experiments to assess the vasodilator effect of selexipag on dissected pulmonary arterial vessel rings in the poster. The work about the vasodilator effect of selexipag has been already published as indicated below. Therefore, she is not included as an author in this study.

Fuchikami C, Murakami K, Tajima K, Homan J, Kosugi K, Kuramoto K, Oka M, Kuwano K. 

A comparison of vasodilation mode among selexipag (NS-304; [2-{4-[(5,6- diphenylpyrazin-2-yl)(isopropyl)amino]butoxy}-N-(methylsulfonyl) acetamide]), its active metabolite MRE-269 and various prostacyclin receptor agonists in rat, porcine and human pulmonary arteries.

 Eur J Pharmacol. 2017;795: 75–83. doi:10.1016/j.ejphar.2016.11.05

＊Figure 3. What is known about the effect of selexipag on RV systolic and diastolic function in SuHx exposed rats ? The authors should provide the (blinded) RVEDP measurements, RV dp/dt max and min. It would be greta if the authors had echo data but it seems it would require the entire study if they do not have those data.

(Ans.) We have not measured the effect of selexipag on RV systolic and diastolic function, because we do not have any diagnostic machines such as ultrasound echocardiography or MRI. We agree that the effect of selexipag on RV systolic and diastolic function should be investigated in a future study.

＊Figure 4A, 6A, 8A – include the name of the staining in the figure (as in Figure 7).

(Ans.) We have added the name of the staining in the figure.

＊Figure 6B – 1st column can’t be empty as you performed statistics on it.

(Ans.) As previously reported by Shinohara et al., occlusive lesions were not observed in pulmonary vessels in normal rats. Thus, the percentages of all normal group were 0%. We performed statistical analysis between the normal group (0%) and the SuHx groups.

Shinohara T, Sawada H, Otsuki S, Yodoya N, Kato T, Ohashi H, et al. 

Macitentan reverses early obstructive pulmonary vasculopathy in rats: early intervention in overcoming the survivin-mediated resistance to apoptosis. 

Am J Physiol Lung Cell Mol Physiol. 2015;308: L523-38. doi:10.1152/ajplung.00129.2014

＊Figure 7. As far as I know, it was not know that selecipag ca induce apoptosis in the media (TUNEL assay), probably SMC. Can this not simpy explained by the supraphysiological doses ? Can the authors provide cell culture experimental data that underpins this finding with much lower selexipag’s active metabolite in culture, in the dose range of 10nM – 1uM ?

(Ans.) We have not assessed the proapoptotic and antiproliferative effect of selexipag and its active metabolite, MRE-269, on pulmonary artery endothelial cells or smooth muscle cells from SuHx rats. However, hemodynamic unloading by pulmonary artery banding and reducing blood flow in the lungs reverses occlusive vascular lesions in the SuHx model [Abe et al., 2016]. This suggests that a strong reduction in hemodynamic stress by pulmonary vasodilation may improve pulmonary vascular remodeling. We mention this in the revised discussion section (lines 567-571).

Abe K, Shinoda M, Tanaka M, Kuwabara Y, Yoshida K, Hirooka Y, et al. 

Haemodynamic unloading reverses occlusive vascular lesions in severe pulmonary hypertension. Cardiovasc Res. 2016;111: 16–25. doi:10.1093/cvr/cvw070

＊Overall, I feel the figures could be also in panels so that the total number of figures can be reduced to approx.. 7, but leave this to the discretion of the handling editor.

(Ans.) According to submission guidelines, the maximum height of the figures is 2625 pixels at 300 dpi. So we separated the results of hemodynamics and vascular remodeling in SuHx rats into two figures in order to keep the resolution of the images and make them easy to see. However, we will follow the instructions of editor about the figures.

＊Have the authors performed any expression studies on the RV and LV, or even only whole lung, to get a handle on the RNA/protein changes induced by selexipag and its active metabolite…. ?

＊Have the authors performed any expression studies on other organs, such as liver and idney that are probably also affected by the demonstrated systemic to suprasystemic RV and PA pressures in SuHx rats ?

(Ans.) We have not measured the effect of selexipag on RNA or protein expression in lung or other organs. We agree that the effect of selexipag on RNA or protein expression should be investigated in a future study.

Minor points:

1. In the introduction after approval (lines 69-70), it is important to address the use of selexigag in pediatric patients (Geerdink et al, Pulm Circ 2017; Koo et al., Cardiol Young, 2019; Rothman et al, Pulm Circ 2020; Hansmann et al, J H Lung Transplant 2020).

(Ans.) We totally agree that the beneficial effect of selexipag in pediatric PAH patients has been reported by several groups, and it is an important observation to improve the treatment of pediatric patients. However, at present, selexipag has not been approved for the treatment of pediatric PAH patients in any country. We are concerned that describing the use of selexipag in pediatric patients could mislead readers into thinking that selexipag is also approved or recommended for the treatment of pediatric patients. This is an ethical issue because Nippon Shinyaku Co. Ltd., which we are employed by, is the manufacturer of selexipag. Therefore, we are afraid that we should not address the use of selexipag in pediatric patients in this paper.

2. There is data not included in the figues in the discussion (lines 532-536). Please, include in the figures/supplemental figures.

(Ans.) We added this data as supplemental figure 1.

3. Discussion line 568: “Selexipag partially reversed RV hypertrophy” – you haven’t shown a direct effect of selexipag on the heart or isolated cardiomyocytes, also as was shown in Legchenko et al.,Sci Trans Med, 2018, at 3+1 weeks, RV diameter is not changed in Sprague Dawley rats, so at the time of administration of selexipag, the RV was not hypertrophied. Needs to be re-written as: administration of selexipag attenuated the development of RV hypertrophy.

(Ans.) According to your suggestion, we have rewritten the sentence in the discussion section (line 617).

 

Reviewer #2: Honda et al. investigated the effects of selexipag in sugen/hypoxia induced PAH. Selexipag has acute vasodilator effect and improved RVSP, RVH and pulmonary artery remodeling in prevention and treatment protocol. This study is interested; however, some concerns are included.

We deeply appreciate your kind advice on our manuscript. Accepting your criticism, the manuscript was revised to make our conclusions more convincing. Hereafter, we will reply to the points you raised.

1. What caused the reverse pulmonary artery remodeling? Could reduction of shear stress by vasodilation of selexipag reverse pulmonary artery remodeling? Or Did selexipag directly affect pulmonary arteries?

(Ans.) Selexipag is a potent prostacyclin receptor agonist and it significantly reduced RVSP in the U46619-induced acute PAH rat model. Thus, the reduction of shear stress by the vasodilator effect of selexipag is probably the primary mechanism of action of the improvement of pulmonary artery remodeling. The antiproliferative effect of selexipag on occlusive vessels in SuHx rats has not been clearly elucidated yet, but it may contribute to the improvement of vascular remodeling. We added a sentence about this point in the discussion section (lines 604-605).

2. Authors should emphasize the novelty of this study. Similar studies had already published.

(Ans.) This is the first report to show that activation of the prostacyclin receptor exerts therapeutic effects on not only the hemodynamics but also on occlusive vascular remodeling and mortality by RV failure in SuHx rat models. We added a description of our findings in abstract/introduction/discussion sections to emphasize the novelty of this study (lines 50-55, 99-109 and 626-635).

3. Did authors examine the proapoptotic and antiproliferative effects in pulmonary artery endothelial or smooth muscle cells from sugen/hypoxia rats?

(Ans.) We have not assessed the proapoptotic and antiproliferative effect of selexipag on pulmonary artery endothelial cells or smooth muscle cells from SuHx rats. We agree that this point should be investigated in a future study.

4. How many dose of selexipag 30mg/kg is in human setting?

(Ans.) Selexipag significantly improved the hemodynamics and RV hypertrophy at doses of 20–60 mg/kg/day in SD SuHx rats, whereas the approved clinical dose for patients with PAH is up to 3.2 mg/day. The AUC0–∞ of MRE-269 after oral administration of 60 mg/kg/day selexipag in rats is about 100-fold higher than the AUC0–∞ of MRE-269 after oral administration of 3.2 mg selexipag in healthy humans [Asaki et al., 2015, Kaufman et al., 2015]. The difference in the dose setting of selexipag between the rat model and human patients is partly because of the species differences in the pharmacokinetics and the binding affinity of selexipag for the prostacyclin receptor. Thus, the binding affinity of MRE-269 for the human prostacyclin receptor is about 10-fold higher than for the rat prostacyclin receptor [Kuwano et al., 2007]. In addition, treprostinil improves the hemodynamics of an SuHx rat model at a dose over 10-fold higher than that used in the clinical or monocrotaline-induced PAH rat model [Chaudhary et al., 2018]. The difference in the dose required to show effectiveness may be due to differences in the severity of PAH in the models. We added mention of these points in an extra paragraph in the discussion section (lines 537-552).

---

## [Decision Letter · Decision Letter 1]

1 Oct 2020

The selective PGI2 receptor agonist selexipag ameliorates Sugen 5416/hypoxia-induced pulmonary arterial hypertension in rats

PONE-D-20-17665R1

Dear Dr. Honda,

We’re pleased to inform you that your manuscript has been judged scientifically suitable for publication and will be formally accepted for publication once it meets all outstanding technical requirements.

Kind regards,

Michael Bader

Academic Editor

PLOS ONE

Additional Editor Comments (optional):

Reviewers' comments:

Reviewer's Responses to Questions

**Comments to the Author**

1. If the authors have adequately addressed your comments raised in a previous round of review and you feel that this manuscript is now acceptable for publication, you may indicate that here to bypass the “Comments to the Author” section, enter your conflict of interest statement in the “Confidential to Editor” section, and submit your "Accept" recommendation.

Reviewer #1: All comments have been addressed

Reviewer #2: All comments have been addressed

2. Is the manuscript technically sound, and do the data support the conclusions?

Reviewer #1: Yes

Reviewer #2: Yes

3. Has the statistical analysis been performed appropriately and rigorously? 

Reviewer #1: Yes

Reviewer #2: Yes

4. Have the authors made all data underlying the findings in their manuscript fully available?

Reviewer #1: Yes

Reviewer #2: Yes

5. Is the manuscript presented in an intelligible fashion and written in standard English?

Reviewer #1: Yes

Reviewer #2: Yes

6. Review Comments to the Author

Reviewer #1: Honda et al. have well addressed the majority of my concerns. Unfortunately, in the revision they haven’t performed new experiments. However, given the current circumstances, I fully understand that it is challenging, costly and time-consuming to perform the new animal experiments or to find collaborators to do the studies that one’s lab is unable to do.

Reviewer #2: (No Response)

7. PLOS authors have the option to publish the peer review history of their article (what does this mean?). If published, this will include your full peer review and any attached files.

Reviewer #1: No

Reviewer #2: No

---

## [Editor Report · Acceptance letter]

5 Oct 2020

PONE-D-20-17665R1 

The selective PGI_2_ receptor agonist selexipag ameliorates Sugen 5416/hypoxia-induced pulmonary arterial hypertension in rats 

Dear Dr. Honda:

I'm pleased to inform you that your manuscript has been deemed suitable for publication in PLOS ONE. Congratulations! Your manuscript is now with our production department. 

Kind regards, 

on behalf of

Prof. Michael Bader 

Academic Editor

PLOS ONE